# Non-solvent post-modifications with volatile reagents for remarkably porous ketone functionalized polymers of intrinsic microporosity

Sirinapa Wongwilawan[1,2], Thien S. Nguyen[3,4,5], Thi Phuong Nga Nguyen[3], Abdulhadi Alhaji[4], Wonki Lim[1], Yeongran Hong[1], Jin Su Park[1], Mert Atilhan[6], Bumjoon J. Kim [1], Mohamed Eddaoudi [4] & Cafer T. Yavuz [1,3,4,5] ✉

Chemical modifications of porous materials almost always result in loss of structural integrity, porosity, solubility, or stability. Previous attempts, so far, have not allowed any promising trend to unravel, perhaps because of the complexity of porous network frameworks. But the soluble porous polymers, the polymers of intrinsic microporosity, provide an excellent platform to develop a universal strategy for effective modification of functional groups for current demands in advanced applications. Here, we report complete transformation of PIM-1 nitriles into four previously inaccessible functional groups – ketones, alcohols, imines, and hydrazones – in a single step using volatile reagents and through a counter-intuitive non-solvent approach that enables surface area preservation. The modifications are simple, scalable, reproducible, and give record surface areas for modified PIM-1s despite at times having to pass up to two consecutive post-synthetic transformations. This unconventional dual-mode strategy offers valuable directions for chemical modification of porous materials.

Porous materials are at the forefront of materials development since they offer superior contact surfaces enabling substantially higher reactivity with substrates through internal voids, while retaining bulk properties of common materials with similar compositions. Porous polymers emerged as a robust but lightweight family of organic structures that provide significant promise in gas separations, sensors, water treatment, and catalysis[1–5]. Among the vast number of examples, polymer of intrinsic microporosity-1 (PIM-1) is a prominent, ladder-type, soluble, and permanently porous polymer[6] with a Brunauer–Emmett–Teller

(BET) surface area of 760–850 m² g⁻¹. Owing to the deficient chain orientation from a twisted core structure, it inherits a combined benefit of a large surface area porous material and the versatility of a linear polymer[7]. Despite the numerous uses of PIM-1, limitations in its functionality and solvent compatibility have led to attempts to modify its chemistry without sacrificing the surface area and accessible pores[2,8–19].

The amidoxime conversion reported by our group[8] and the amidation method by others[9,10] are noteworthy strategies in that these successful chemical modifications minimized the loss of surface area

[1]Department of Chemical and Biomolecular Engineering, Korea Advanced Institute of Science and Technology (KAIST), 291 Daehak-ro, Yuseong-gu, Daejeon 34141, Republic of Korea. [2]PTT Global Chemical Public Company Limited, Bangkok 10900, Thailand. [3]Oxide & Organic Nanomaterials for Energy & Environment Laboratory, Physical Science & Engineering (PSE), King Abdullah University of Science and Technology (KAUST), Thuwal 23955, Saudi Arabia. [4]Advanced Membranes & Porous Materials Center, PSE, KAUST, Thuwal 23955, Saudi Arabia. [5]KAUST Catalysis Center, PSE, KAUST, Thuwal 23955, Saudi Arabia. [6]Department of Chemical and Paper Engineering, Western Michigan University, Kalamazoo, MI 49008-5462, USA. ✉e-mail: cafer.yavuz@kaust.edu.sa

and porosity. The surface area retention of our non-invasive amidoximation method has also been extended to the modification of insoluble network polymers, which featured the conversion of a covalent organic polymer, COP-122, into amidoxime COP-122 (COP-122-AO)[4]. In particular, amidoxime PIM-1 (AO-PIM-1) was studied extensively because of its exceptional solubility in polar solvents, leading to promising applications in gas separation membranes[20–26].

The amide PIM-1 formation is also remarkably effective. In 2014, Satilmis et al.[9] transformed PIM-1 with a 10% NaOH solution ($H_2O$/ EtOH = 1/1 (w/w)) at 100 °C to achieve 418 $m^2\ g^{-1}$ at 89% conversion, while, two years later, Yanaranop et al.[10] utilized volatile $H_2O_2$ (25%) in a DMSO solution. With a semi-volatile system, the latter chemistry yielded a higher surface area of 527 $m^2\ g^{-1}$, and afforded nearly quantitative conversion. However, none of the reports, including us, made a note of the roles and advantages of volatile reagents.

Indeed, these examples evidently imply the importance of rapid elimination of the reagents in a post-synthetic treatment to accomplish porosity conservation. In addition, volatile substances could also act as porogens, effectively preventing undesired collapse in the irregularly packed linkages. However, one challenge with volatile reagents is that they evaporate quickly, especially at elevated temperatures. Therefore, even if excess amounts are usually employed, the conversions might still not reach quantitative levels. To alleviate this issue, an unconventional approach that could reduce porosity loss by facilitating efficient pore structuring with high reaction conversions is direly needed.

In order to make an attempt to improve conversions, we decided to focus on how we carry out the modifications. Traditional reactions usually involve the complete mixing of reagents in a solvent for a uniform dissemination of reactive components (Fig. 1). However, in such homogeneous mixtures of soluble porous organic polymers, the necessary molecular collisions are truly hindered because of the dominant presence of the solvents. In addition, fully solvated substances would not be affected by the positive driving force stemming from additional buoyancy from surface tension variations[27]. The concentration gradients will also be lost due to effective mixing, removing the possibility of densely localized reactive media. Volatile reagents would also be better stabilized in a compatible solvent mixture, withdrawing the chances of a fast evolution driven reaction kinetics. The solubility of PIMs that derived from the interaction between polymer chain of PIMs and solvent becomes a barrier, limiting the reaction between PIMs and modifying reagent. In contrast, mechanochemical, solventless grinding approaches can bring about rapid reactions due to the maximal contact of the reactants, but both of these methods best operate with non-volatile substances[28–30]. Therefore, a compromise that

implements volatile reagents in a medium that facilitates maximum intermolecular interaction would be a promising answer. In such a design, a volatile modifying agent would be dissolved in a low boiling point solvent in which the reacting counterpart, the porous material, is not soluble (Fig. 1). Such an approach could be called a non-solvent approach.

Here, we propose a non-solvent approach to achieve record high porosity retention in a series of modified PIM-1s with previously inaccessible reactive functionalities. This work marks an in-depth investigation of the effect of volatile and non-volatile reagents on the porosity of modified PIM-1s. Using the non-solvent approach, we were able to transform PIM-1 into an even more microporous ketone-functionalized PIM-1 (Ketone-PIM-1 or in short K-PIM-1). The resulting carbonyl electrophile of K-PIM-1 was effectively converted into numerous functional groups, including alcohols, imines, and hydrazones, each of which were not previously reported for PIM-1 chemistry. The non-invasive nature of the protocol was reinforced by the porosity conservation observed in the product of K-PIM-1 and the volatile methylamine reaction (Methylimine-PIM-1, denoted as MI-PIM-1, with the highest surface area and micropore content of a PIM-1 derivative to date). In addition, the probe gas uptake of the resulting materials was substantially improved. MI-PIM-1 posed a significant hydrogen uptake performance while the covalently tethered PEI-PIM-1 showed remarkably high $CO_2$ uptake and selectivity over $N_2$, a pre-eminent requirement for carbon dioxide capture.

## Results

### Synthesis and characterization with non-solvent approach

In a typical, well-established synthesis of PIM-1[31], large quantities of 5,5',6,6'-tetrahydroxy-3,3,3',3'-tetramethyl-1,1'-spirobisindane (TTSBI) were allowed to react with equimolar tetrafluoroterephthalonitrile (TFTPN) in the presence of an excess base. Depending on the temperature of the reaction, the solvent choice differed, where the high temperature (HT, 160 °C) reflux used dimethylacetamide (DMAc) and toluene mixture[32,33] and the low temperature (LT, 65 °C) protocol employed dimethylformamide (DMF)[8,31]. The HT procedure provided rapid product formation but with much lower molecular weight. The LT procedure, therefore, was apparently more suitable when film making was required (Supplementary Table 1). Consequently, PIM-1 derived from the LT procedure was chosen for the post-synthesis treatments. The characterization of PIM-1 followed common techniques such as GPC, BET, NMR, FT-IR, EA, and XPS (see Methods section and Supplementary Information for details).

To make K-PIM-1, an excess amount of a commercial, volatile reagent methylmagnesium bromide ($CH_3MgBr$) in different solvents

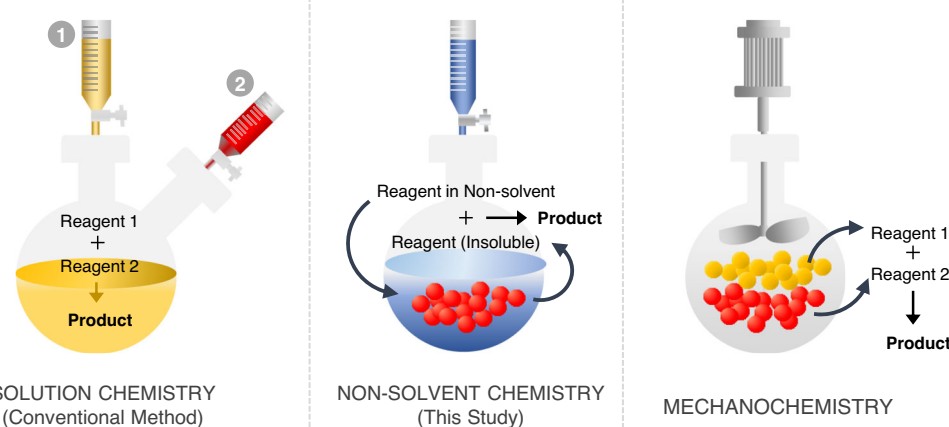

**Fig. 1 | Conceptual representation of available chemical modifications for PIMs.** Synthesis options for a successful modification of soluble porous materials without losing porous properties: conventional synthetic method (left), solventless mechanochemistry (right), and our proposed non-solvent approach (middle).

**Fig. 2 | The quantitative post-modification of PIM-1 through conventional and non-solvent methods with a volatile Grignard reagent, methylmagnesium bromide (CH₃MgBr).** Ketone-PIM−1 (K-PIM-1) was further reacted by the same reagent to make Alcohol-PIM−1 (OH-PIM-1) with remarkably high retention of porosity, considering the two-step reaction. Solvent: (i) CH₃MgBr in tetra-hydrofuran (THF), THF solvent at 0 °C → RT for 24 h (ii) 0.5 M hydrochloric acid

(HCl) in methanol (MeOH), H₂O at 60 °C for 4 h. Non-solvent: (i) CH₃MgBr in diethyl ether (Et₂O) at 0 °C → RT for 24 h (ii) 0.5 M HCl in MeOH, H₂O at 60 °C for 4 h. K-PIM−1 was further functionalized by phenylhydrazine (PhNHNH₂) and poly-ethylenimine (PEI) to show versatility in the reactive portfolio and to create CO₂ adsorbents.

was used: (i) tetrahydrofuran (THF) for a conventional approach and (ii) diethyl ether (Et₂O) for the new non-solvent approach (Fig. 2). K-PIM-1 products from both synthesis routes were then treated one more time with the same CH₃MgBr solutions to afford the corresponding alcohol-PIM-1s (named as OH-PIM-1). The existence of ketone functionality in the targeted K-PIM-1 was first qualitatively confirmed by its reaction with hydrazine under a mild acidic condition to form hydrazone-PIM-1 (or in short, HZ-PIM-1). In order to enhance the hydrogen uptake and the selectivity towards CO₂ gas, the different amine moieties were subsequently introduced in K-PIM-1 by using the highly volatile methylamine and the bulky, non-volatile poly-ethylenimine (PEI), respectively. All chemical transformations were thoroughly assessed by FT-IR spectroscopy, elemental analysis (EA), solid-state cross-polarization magic angle spinning (CP-MAS) ¹³C NMR, and X-ray photoelectron spectroscopy (XPS) (Fig. 3a, Table 1, and Supplementary Figs. 6–8). In addition, K-PIM-1 showed excellent batch-to-batch reproducibility both in porosity and in percent conversion (Supplementary Table 2).

The most direct analysis of a successful transformation of PIM-1 is to monitor the nitrile peak in a FT-IR spectrum[8]. K-PIM-1 from the non-solvent method showed a clear absence of the nitrile peak (Fig. 3a), -C≡N stretching around 2239 cm⁻¹, and the presence of a clearly strong peak for C = O at 1715 cm⁻¹, suggesting that the cyano group had been converted to ketone functionality. The mechanism for the ketone formation can be found in Supplementary Fig. 9. A similar transfor-mation was observed by the conventional method, as expected (Sup-plementary Fig. 10). The further reaction of K-PIM−1 with a Grignard reagent to yield OH-PIM-1 appeared to be relatively straightforward. The broad peak of O-H stretching was visible at 3597 cm⁻¹ with the decrease of carbonyl peak, implying that OH-PIM-1 contained a mix-ture of both hydroxyl and carbonyl functionalities. The hydrazone formation also occurred readily. The reaction between K-PIM-1 and phenylhydrazine yielded a distinct orange powder (HZ-PIM-1). More-over, it can be seen from Fig. 3a that the FT-IR spectrum of HZ-PIM-1 shows a significant decrease of C = O at 1715 cm⁻¹ with the increase of a noticeable peak at 1598 cm⁻¹ associated with C = N stretching in com-parison with K-PIM-1, emphasizing the success of the hydrazone for-mation. This evidence confirmed the establishment of the ketone

functionality in the PIM-1 structure. In addition, K-PIM-1 was analyzed by ICP-MS to detect any metal trace trapped in the pores of the final product. It is clear that magnesium salt was not retained in the pores of K-PIM-1 after the post-modification process (Supplementary Table 3).

In the successive functionalizations, the attachment of methyla-mine and polyethylenimine (PEI) to K-PIM-1 greatly diminished the signal of C = O stretching at 1715 cm⁻¹ in FT-IR, whereas there remained some residual ketone contents in MI-PIM-1 owing to the reversibility of the reaction of the volatile amine. The distinctive imine C = N stretching signal of MI-PIM-1 could be found at 1659 cm⁻¹, while that of PEI-PIM-1 appeared at 1656 cm⁻¹. In addition, the vibration at around 3300–3500 cm⁻¹ was observed in PEI-PIM-1, which featured the N-H stretching mode. Moreover, the broad peak at 1650-1580 cm⁻¹ indi-cated N-H bending vibration.

To demonstrate the versatility of the ketone group in K-PIM-1 for chemically anchoring with amines, PIM-1 was allowed to directly react with phenylhydrazine and PEI under identical conditions. As expected, FT-IR spectra revealed no change in starting materials. This clearly demonstrates that without a carbonyl group, the pristine PIM-1 is incapable of reacting with amines except if under a specific condition such as heat treatment (Supplementary Fig. 11)[34].

For further identification of chemical structure, solid-state ¹³C CP-MAS NMR analysis showed the important characteristic peaks (Sup-plementary Fig. 6). For ¹³C-NMR of K-PIM-1, it revealed the distinct chemical shift at 193 ppm which is attributed to the carbonyl carbons. This peak becomes diminished after second post-modifications were performed, confirming that the ketone functionality was successfully converted to the desired functional groups.

Elemental analysis results granted firm support to the anticipated chemical structures as they agreed well with the theoretical calculation (Table 1). Those results also correlated well with XPS survey scan (Supplementary Figs. 7, 8). The nitrogen content in K-PIM-1 (non-sol-vent) is almost undetectable, as corroborated by N 1s XPS spectra (Supplementary Fig. 12). A closer examination (Supplementary Table 4) revealed that the degrees of conversion for K-PIM-1 and OH-PIM-1 via the non-solvent approach were superior to those via the solvent method. The K-PIM−1 formation via the non-solvent process achieved more than 90% conversion, whereas that by the solvent

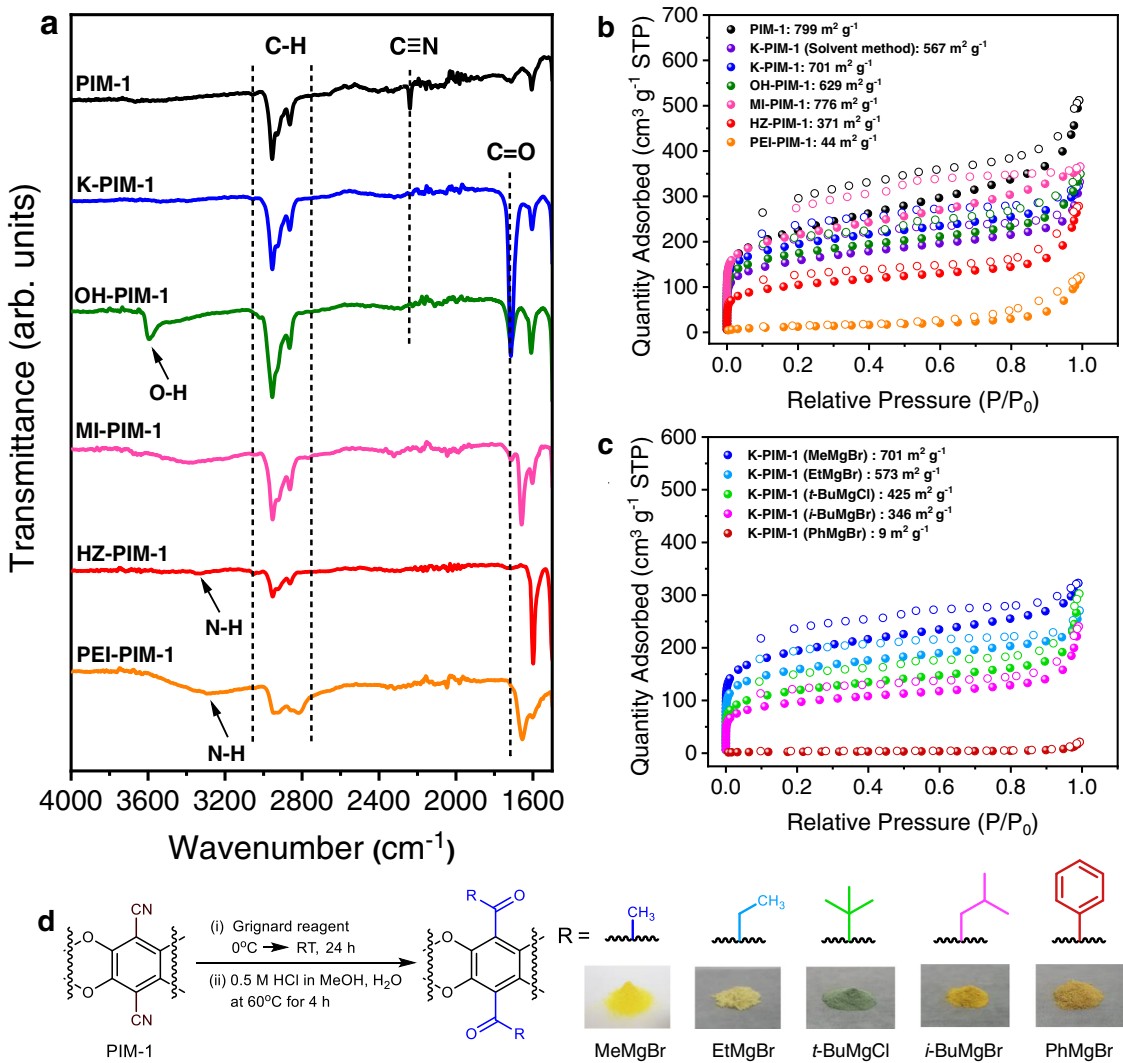

**Fig. 3 | Characterization and volatile reagent effects in the non-solvent post-modification. a** FT-IR spectra and **b** N₂ adsorption/desorption isotherms measured at 77 K. **c** Comparison of N₂ adsorption/desorption isotherms and BET surface areas for post-modifications of PIM-1 by volatile and non-volatile reagents via the non-solvent method. and **d** Schematic illustration of the post-synthetic modification by using different modifying substances. The final products are methyl K-PIM−1 (MeMgBr), ethyl K-PIM-1 (EtMgBr), tert-butyl K-PIM-1 (*t*-BuMgCl), isobutyl K-PIM-1 (*i*-BuMgBr), and phenyl K-PIM-1 (PhMgBr), which derived from the reaction between PIM-1 and Grignard reagents; methylmagnesium bromide, ethylmagnesium bromide, tert-buthylmagnesium chloride, isobuthylmagnesium bromide, and phenylmagnesium bromide, respectively.

method, it was approximately 55.9%. The elemental composition obtained by XPS survey scan was analyzed to confirm the validity of the degrees of conversion. As can be seen in Supplementary Table 5, the conversions derived from XPS analysis were nearly the same as EA results. It is evident from these results that the non-solvent approach was more powerful in terms of reactivity.

Thermal stabilities of the synthesized products were assessed by thermogravimetric analysis (TGA) measurements which were carried out under air and inert atmospheres (Supplementary Fig. 13a, b). The modified PIMs were slightly less thermally stable than the parent PIM −1. The main decomposition onsets of all derivatives were greater than 300 °C, where the initial mass loss at around 100 °C was presumably caused by the removal of trapped moisture or volatile organics. From the stability measurements, we confirm that the inclusion of amines into the pores of PIM by covalent bonding shows exceptionally higher thermal stability than the physical impregnation[35].

## Volatile reagent effect in non-solvent post-modifications

One advantage of Grignard reagents is that the alkyl group bulkiness of the nucleophilic carbon can be tuned. This allows for systematic

evaluation of the steric effects, and in our case, the volatility of the reagent. To investigate the effect of volatility in K-PIM-1 formation, we used the same batch of PIM-1 to treat with methylmagnesium bromide (MeMgBr), ethylmagnesium bromide (EtMgBr), isobutylmagnesium bromide (*i*-BuMgBr) and phenylmagnesium bromide (PhMgBr) in identical non-solvent conditions (Fig. 3d). The products were analyzed by FT-IR, elemental analysis (EA), and XPS as shown in Supplementary Figs. 14−16 and Supplementary Tables 5, 6 to confirm the conversion. It was obviously seen that ketone formation of all derivatives was facilely achieved with satisfied conversion yield (>80%). To deeply observe the advantage of volatile reagent on porosity preservation, N₂ adsorption/desorption isotherms and the apparent surface areas were evaluated (Fig. 3c, Supplementary Figs. 17−19, and Supplementary Table 6). So, based on the same series of Grignard reagents, the modifying group was orderly varied from the highly volatile to non-volatile as follows: methyl>ethyl>isobutyl>phenyl. As expected, PIM-1 modified by volatile reagents: MeMgBr, EtMgBr, and *i*-BuMgBr showed high surface areas (SA_BET: 701 m² g⁻¹, 573 m² g⁻¹, and 346 m² g⁻¹, respectively). In comparison, phenyl ketone product from the non-volatile PhMgBr reagent came out to have negligible surface area. This indicates that

**Table 1 | Gas uptake properties and elemental (CHNO) analyses of PIM-1 and derivatives**

| Material | $SA_{BET}$ [m² g⁻¹] | $SA_{Langmuir}$ [m2 g⁻¹] | $SA_{micro}$ [m² g⁻¹] | Micro vol. [cm³ g⁻¹] | Toal Pore vol. [cm³ g⁻¹] | Elemental analysis %exp [Theo.] | | | | $CO_2$ uptake at 1 bar [mmol g⁻¹] | | | $CO_2$ $Q_{st,max}$ at 0.05 bar [kJ mol⁻¹] | $CO_2/N_2$ Gas Selectivity [15:85] at 298 K |
|---|---|---|---|---|---|---|---|---|---|---|---|---|---|---|
| | | | | | | C | H | N | O | 273 K | 298 K | 323 K | | |
| PIM-1 | 799 | 939 | 277 | 0.25 | 0.74 | 74.6 [75.6] | 4.3 [4.4] | 6.3 [6.1] | 13.1 [13.9] | 2.40 | 1.31 | 0.83 | 28.9 | 27 |
| K-PIM-1 (Solvent) | 567 | 668 | 257 | 0.20 | 0.48 | 73.5 [75.3] | 5.0 [5.3] | 2.8 [0] | 15.5 [19.4] | 1.63 | 0.96 | 0.57 | 28.7 | 42 |
| K-PIM-1 | 701 | 822 | 366 | 0.24 | 0.46 | 73.9 [75.3] | 5.2 [5.3] | 0.4 [0] | 18.1 [19.4] | 2.14 | 1.22 | 0.64 | 30.1 | 47 |
| OH-PIM-1 | 629 | 752 | 329 | 0.21 | 0.49 | 74.1 [75.3] | 6.1 [6.5] | 0.2 [0] | 17.7 [18.2] | 1.85 | 1.02 | 0.52 | 27.2 | 70 |
| MI-PIM-1 | 776 | 893 | 389 | 0.27 | 0.54 | 74.9 [76.1] | 5.9 [6.2] | 4.9 [5.4] | 11.3 [12.3] | 2.24 | 1.28 | 0.70 | 26.9 | 18 |
| HZ-PIM-1 | 371 | 421 | 154 | 0.12 | 0.40 | 75.6 [76.5] | 5.6 [5.7] | 7.7 [8.3] | 9.9 [9.5] | 1.00 | 0.54 | 0.26 | 19.5 | 53 |
| PEI-PIM-1[a] | 44 | 60 | – | 0.01 | 0.18 | 67.0 | 6.9 | 10.7 | 10.8 | 2.34 | 1.70 | 1.17 | 44.5[b] | 316 |

[a]The theoretical value for elemental composition of PEI PIM–1 was not calculated owing to the structural complexity of polyethylenimine dendrimers.
[b]$Q_{st-max}$ under $CO_2$ loading (mmol g⁻¹) at 0.90 bar.

the surface area significantly depends on the volatility of modifying group. Highly volatile reagents yielded the highest surface area. Further investigation was thoroughly performed by using tert-butylmagnesium chloride (*t*-BuMgCl), a bulky volatile reagent. Surprisingly, the bulky volatile reagent of *t*-BuMgCl could maintain the surface area (425 m² g⁻¹), even though bulky groups usually hinder the porosity of porous polymers by pore filling. Since the full conversion to K-PIM-1 by *t*-BuMgCl was not achieved due to steric hindrance, half conversion of a non-volatile K-PIM-1 (PhMgBr) was additionally performed to make a fair comparison and examine the porosity retention of the final product. It was clearly seen that K-PIM-1 (PhMgBr) at half conversion was also virtually non-porous ($SA_{BET}$: 9 m² g⁻¹) (Fig. 3c and Supplementary Figs. 20, 21). This dramatic difference in surface areas provides a clear demonstration of the effect of the volatility for modifying agents on the surface area during post-modification treatments.

## Surface area and porosity

The surface areas of all PIM-1s and derivatives were determined using BET theory on $N_2$ adsorption-desorption isotherms at 77 K (Fig. 3b and Supplementary Figs. 17, 22–27). The calculations included determination of the ideal pressure range through Rouquerol plots (detailed calculations can be found in section 4 of the supplementary information). We have used best fitting values as it is well known that BET areas could deviate significantly just by simple data point exclusions[36,37]. The results revealed the non-invasive nature of our method. Notably, the outcome of the non-solvent approach occurred to be superior by not only giving higher conversions but also significantly uplifting the surface areas of K-PIM−1 and OH-PIM-1 from 567 to 701 m² g⁻¹ and 460 to 629 m² g⁻¹, respectively, compared to the conventional solvent method (Supplementary Table 7). Interestingly, applying a well-documented alcohol treatment method for pore cleansing[10,12,38] did not result in substantial enhancement compared to the impact of the non-solvent method (Supplementary Fig. 28).

The slight reduction of the surface area going from K-PIM-1 to OH-PIM-1 could be ascribed to the effect of intermolecular interactions of the formed hydroxyl groups, which induce hydrogen bonding between neighboring chains and subsequently narrow the pore cavity[8,12]. As expected, the incorporation of different amine pendant groups varied the resulting surface areas. For example, post-synthetically modifying K-PIM-1 with methylamine, a highly volatile substance, led to the highest surface area ($SA_{BET}$ of MI-PIM-1: 776 m² g⁻¹) among all amine modifiers while the surface area of HZ-PIM-1 (371 m² g⁻¹) and PEI-PIM-1 (44 m² g⁻¹) were tuned down in accordance with the increase in the respective amine sizes (Fig. 3b). All derivatives except PEI-PIM-1 showed a Type I $N_2$ sorption isotherm with slight behavior of a Type IV, representing combination of micro-mesoporosity[39–41]. They had a sizeable $N_2$ uptake at low relative pressures. Their long hysteresis loop expanding down to low $P/P_0$ was also observed, which could be explained by many possible reasons such as pore network effects, the swelling effect, and diffusional limitations by pore blocking effects[6,42,43]. In contrast, PEI-PIM-1 exhibited a Type IV isotherm, suggesting the transformation of micropores to mesopores. The existence of micropores and mesopores is demonstrated by the pore size distribution calculated by non-local density functional theory (NLDFT) applying the carbon slit pore model (Supplementary Fig. 29)[44]. As seen in Table 1, the micropore surface area of MI-PIM-1 (389 m² g⁻¹), K-PIM-1 (366 m² g⁻¹), and OH-PIM-1 (329 m² g⁻¹) were greater than the parent PIM-1 (277 m² g⁻¹), whereas that of HZ-PIM-1 (154 m² g⁻¹) and PEI-PIM-1 emerged to be lower. The origin of the different micropore surface areas could be ascribed to the size of the modifying agents. The functional group that was slightly bulkier than the cyano group would occupy the larger cavities, enabling the creation of a micropore, whereas the even larger functional groups tended to close that available pore, leading to a lower micropore content[8,12]. In light of this

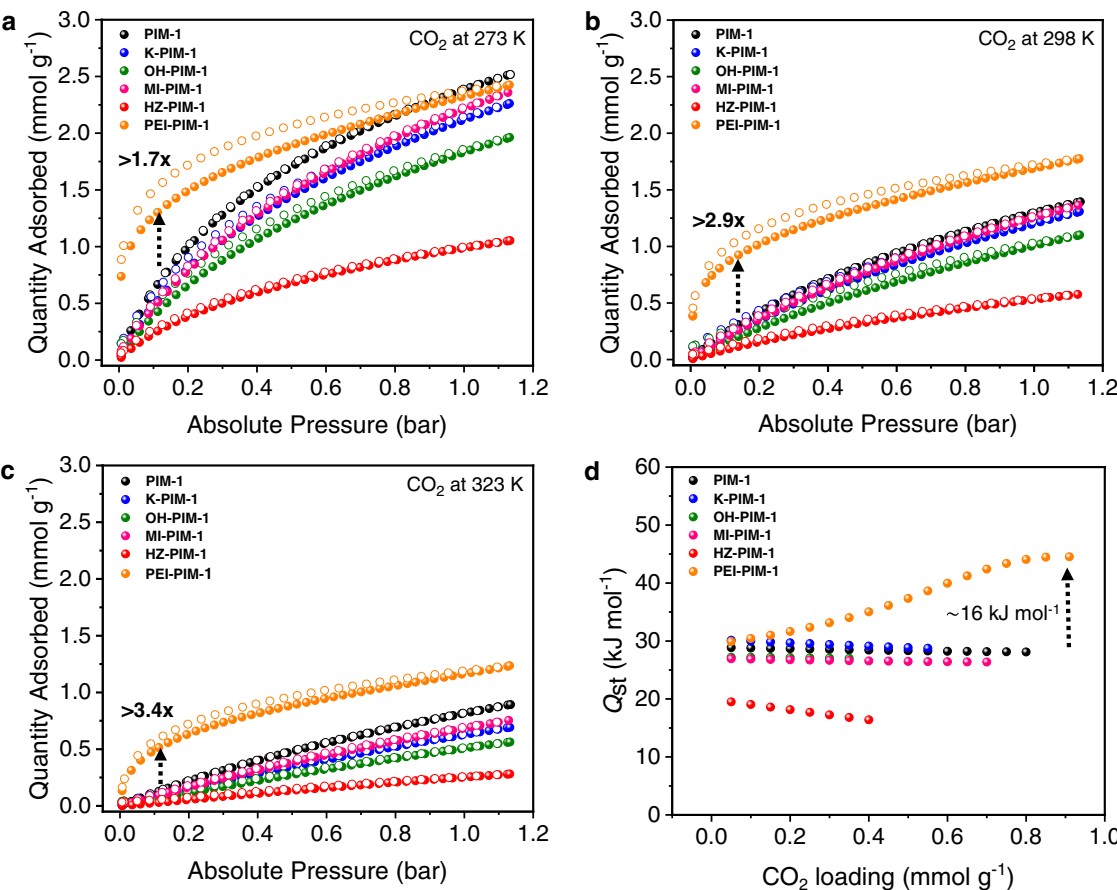

**Fig. 4 | Carbon dioxide capture performances of PIMs.** $CO_2$ uptake isotherms of PIM-1 and derivatives at **a** 273 K **b** 298 K and **c** 323 K, and their corresponding **d** isosteric heat of adsorption ($Q_{st}$) values calculated from the adsorption isotherms at 273 K, 298 K and 323 K.

finding, we allude that the utilization of volatile substances could effectively maintain the specific surface area. Among the reported functionalized PIM-1s to date[2,8-19], the new PIMs introduced in this work (MI-PIM-1, K-PIM-1, and OH-PIM-1) now rank top three in the comparisons table of porosity, despite the fact that MI-PIM-1 and OH-PIM-1 have passed the modification twice (Supplementary Table 8).

**Gas uptake studies**

Amine modified porous materials are often developed for the purpose of $CO_2$ capture and separation since amines provide strong chemisorption for the acidic gases[45-47]. We, therefore, measured the $CO_2$ uptake isotherms of modified PIMs at 273 K, 298 K, and 323 K up to 1 bar (Fig. 4a–c, Methods, and Supplementary Table 9). The $CO_2$ adsorption capacity of PIM-1 derivatives except PEI-PIM-1 were found to be close to or lower than that of the parent PIM-1 at all temperatures due to the decrease of surface area. However, the performance of K-PIM-1 (non-solvent) and MI-PIM-1 was comparable to the pristine PIM-1 as their porosities were well preserved. No hysteresis loops were observed from the isotherms, indicating the characteristic of typical physisorption. On the contrary, PEI-PIM-1 significantly boosted $CO_2$ adsorption at low partial pressures and its isotherm showed a distinct hysteresis loop. At 0.15 bar, $CO_2$ uptake reaches 1.39, 0.94, and 0.57 mmol g$^{-1}$ at 273 K, 298 K, and 323 K respectively, which emerges to be 1.7, 2.9, and 3.4-times higher than the parent PIM-1. The considerable enhancement of the $CO_2$ capacity was achieved due to the strong chemical adsorption brought about by the high density of amine moieties[48]. Considering the $CO_2$ uptake at a low partial pressure (0.15 bar at 298 K) of any PIM base that have been reported to date, the uptake capacity of PEI-PIM-1 is one of the highest, and most certainly the most robust with the chemically tethered PEI (Supplementary

Fig. 30)[8,14,17,18,35,41,49]. Plus, PEI-PIM-1 was comparable to other PEI-tethered materials as shown in the Supplementary Table 10[50-59].

To shed more light on the impact of pendant groups on the chemical interaction, isosteric heat of adsorption ($Q_{st}$) were determined by the dual-site Langmuir-Freundlich fitting of $CO_2$ adsorption isotherms covering three temperatures at 273 K, 298 K, and 323 K using Clausius−Clapeyron equation (See Methods section for details, Supplementary Figs. 31–36, and Supplementary Tables 11, 12)[46,60-62]. The $CO_2$ adsorption enthalpies were plotted as a function of $CO_2$ loading as displayed in Fig. 4d. The maximum isosteric heat of adsorption ($Q_{st,max}$) values of all derivatives except PEI-PIM-1 were held in the range of 19.5−30.1 kJ mol$^{-1}$ (Table 1). As predicted, covalent immobilization of PEI through ketone functionality of K-PIM-1 could tune binding energy into an ideal range ($Q_{st,max}$ = 44.5 kJ mol$^{-1}$). PEI-PIM-1 also showed expansion of $Q_{st}$ at higher $CO_2$ loading. This is due to high favorability of PEI-PIM-1 towards $CO_2$. Upon increasing the $CO_2$ loading, $CO_2$ molecules largely occupied within its structure, enhancing the interactions between adsorbent-adsorbate and adsorbate-adsorbate. At higher coverage, the $CO_2$ molecules can force expansion of the flexible PEI chains, strengthening their binding by additional access to more nitrogens and the ensuing complexation[63]. Ultimately, the interaction between $CO_2$ molecules and the polyamines can be continuously strengthened to increase in $Q_{st}$[64,65]. As a result of PEI inclusion, the $Q_{st}$ value fits into chemisorption and physisorption boundary (40−50 kJ mol$^{-1}$) and, therefore, more favorable as it balances the strong binding interaction and the regeneration energy penalty[46]. During this calculation, we also demonstrated an alternative approach by performing Dual-Site Langmuir-Freundlich (DSLF) model fitting for $CO_2$ adsorption isotherms at 273 K, 298 K, and 323 K via logarithmic-scale plots using OriginPro[66]. The fitted parameters were

also used to calculate $Q_{st}$ value (Supplementary Figs. 37, 38, and Supplementary Tables 13, 14). The experimental data sets were fitted well for both Wolfram Mathematica and OriginPro. The $Q_{st}$ values obtained from both sources revealed the same tendency.

Another key property that can help mitigate the total cost of $CO_2$ capture is a high $CO_2$ selectivity over other gases. Therefore, nitrogen and methane adsorptions were further collected to appraise the industrially relevant $CO_2/N_2$ and $CO_2/CH_4$ selectivities. (Supplementary Figs. 39, 40). The ideal adsorption solution theory (IAST) method was employed as it is a reliable prediction for the actual gas separations[46,60–62,67]. The gas selectivity for a simulated flue gas containing a 15% $CO_2$ over 85% $N_2$ mixture is presented in Table 1, Supplementary Table 9, and Supplementary Fig. 41a-c. As expected, PEI-PIM-1 showed a significant improvement of $CO_2/N_2$ selectivity (>270) at all temperature ranges, compared to other derivatives. This prominent performance is derived from the material's high chemisorption of $CO_2$. Additionally, amine pendant groups accommodated in the pores would negate the adsorption of $N_2$ and at the same time allow for the accessibility of $CO_2$ thanks to the higher polarizability and the characteristic quadrupole[48,68]. On the other hand, the $CO_2/N_2$ selectivity of MI-PIM-1 exhibited lower than that of the parent PIM-1 since the small pores of MI-PIM-1 facilitated a better containment of both probe gases, including $N_2$. To show the versatility of functionalized PIM-1s, we further investigated their performance for $CO_2$ separation from methane, the predominant component of natural gas. Hence, the $CO_2/CH_4$ selectivity of all products was calculated at 50/50 gas mixture. As shown in Supplementary Table 9 and Supplementary Fig. 42, the selectivity of PEI-PIM-1 for $CO_2$ over $CH_4$ was greatly superior (>3000) at 273 K while the selectivity of other derivatives fell in the range 7–9. The exceptionally high $CO_2/CH_4$ selectivity of PEI-PIM-1 was an outcome of the dominant basicity of the nitrogen-rich PEI, providing abundant interaction sites for $CO_2$ via acid-base binding. To further appraise the performance of PEI-PIM-1 compared with PIM-1 in the actual adsorption-based separation process, $CO_2$ breakthrough experiments for PEI-PIM-1 and PIM-1 were tested at ambient temperature using 80.75% $N_2$, 14.25% $CO_2$, 5% He gas mixtures (Supplementary Figs. 43, 44). The dynamic adsorption capacity of PEI-PIM-1 (0.58 mmol $g^{-1}$) was superior to PIM-1 (0.31 mmol $g^{-1}$), which is consistent with the pure gas isotherms (see Supplementary Information section 5 for methods and calculations). The adsorption value obtained from static and dynamic conditions may not be equivalent as it is theoretically known that the former yields higher $CO_2$ uptake than the latter[69,70]. Also, axial dispersion, mass transfer and adsorption kinetics effects take place during the dynamic operation, influencing the shape of breakthrough curve used for calculating the adsorption capacity[71]. Apart from the adsorption capacity, it is important to note that $CO_2$ breakthrough curve of PEI-PIM-1 showed wider breakthrough time gap between $CO_2$ and $N_2$ compared to the parent PIM-1, indicating a substantially better separation performance of $CO_2/N_2$ gas pair.

The hydrogen storage properties for all functionalized materials were also examined up to 1 bar and 77 K (Supplementary Fig. 45 and Supplementary Table 9). The order of PIM-1 derivatives based on $H_2$ uptake performance was MI-PIM-1>PIM-1 > K-PIM-1>OH-PIM-1>HZ-PIM-1>PEI-PIM-1. These results were in good agreement with the common observation that $H_2$ uptake could be increased upon narrowing the pore size and enhancing the dominance of micropores[39,72,73]. The hydrogen uptake of MI-PIM-1 was 10.2% higher than the original PIM-1 due to its smaller pore size. Even though K-PIM-1 had an abundance of small pores, its uptake performance slightly declined owing to the lower micropore volume than that of PIM-1. To the best of our knowledge, MI-PIM-1 is the first material among the post-modified cyano group on PIM-1 that reports a major improvement over the hydrogen storage capacity of the original PIM-1[72,74,75]. Our investigations will continue to explore this unique behavior.

## Solubility and processability
It is widely known that PIM-1 is almost exclusively soluble in tetrahydrofuran (THF) and chloroform ($CHCl_3$)[7]. After the functionalizations carried out in this study, the solubility of PIM-1 derivatives was noticeably changed (Supplementary Table 15). Even though the main modified structure, K-PIM-1, was not perfectly soluble, it is worth mentioning that the processability of K-PIM-1 was not hindered and allowed to proceed through film-to-film conversion (Supplementary Figs. 46, 47). Moreover, even after prolonged storage of over one month, HZ-PIM-1 was found to be highly soluble in liquid amines such as aniline, triethylenetetramine (TETA), piperidine, and quinoline. This observation could be due to a plausible mechanism of dynamic imine chemistry (Supplementary Fig. 48a, b and Supplementary Table 16).

## Discussion
In an effort to modify porous materials without sacrificing porosity, we have arrived at a unique strategy for the chemical modification of PIM-1 by consolidating the idea of using volatile reagents coupled with a non-solvent approach. This conceptual design was proven by a systematic study supported by experimental evidence showing the effect of modification conditions on the porosity of the derivatives. With the reproducible results, our facile synthetic protocol showed a great potential in maintaining fundamental surface properties of the porous polymers with quantitative conversion yields, which overcame the challenging shortcomings of the conventional modification processes that have obstructed materials development. In the light of our experimental findings, we anticipate that this unconventional route could be a promising new direction for the post-synthesis of new porous materials in the future. Furthermore, newly synthesized K-PIM-1 could be extended to be further functionalized by various amine types to accomplish other desired performance goals, such as applications in heterogeneous catalysis.

## Methods
### Materials
Tetrafluoroterephthalonitrile (99%), methylmagnesium bromide solution (3.0 M in diethyl ether), ethylmagnesium bromide solution (3.0 M) in diethyl ether, tert-butylmagnesium chloride (2.0 M) in diethyl ether, phenylmagnesium bromide solution (3.0 M) in diethyl ether, polyethylenimine (Average Mw ~800), phenylhydrazine (97%), methylamine solution (33 wt.% in absolute ethanol), $CDCl_3$ (99.8%), anhydrous lithium chloride (99%), anhydrous potassium carbonate (99.9%), hydrochloric acid (37%), anhydrous ethyl alcohol (200 proof, >99.5%), anhydrous *N,N*-dimethyl formamide (99%), anhydrous toluene (99.8%), anhydrous *N,N*-dimethylacetamide (99.8%) and anhydrous tetrahydrofuran (>99.9%) were obtained from Sigma-Aldrich, USA. Methylmagnesium bromide, a 1.0 M solution in THF and isobutylmagnesium bromide solution in (2.0 M) diethyl ether were supplied by Acros Organics, Belgium. 5,5′,6,6′-tetrahydroxy-3,3,3′,3′-tetramethyl-1,1′-spirobisindane (>96%) was purchased from TCI, Japan. Glacial acetic acid (99.7%), acetone, (99.5%), ethyl alcohol (99.5%), chloroform (99.5%), 1,4-dioxane (99.5%), and methanol (99.8%) were acquired from SAMCHUN, South Korea. All chemicals and solvents were used as received. Potassium carbonate was ground into a fine powder and dried at 120 °C for 24 h before use. CAUTION: Grignard reagents react with acids violently, use extreme caution and ample cooling when quenching the reactions.

### Synthesis of PIM-1
The detailed synthesis of PIM-1 from both low temperature and high temperature methods can be found in Supplementary Methods.

### Synthesis of K-PIM-1 and OH-PIM-1 via solvent method
In solvent approach, PIM-1 powder obtained from low temperature (LT) method (0.46 g, 1.0 mmol) was dissolved in anhydrous THF

(20 mL) and stirred at room temperature under an inert atmosphere of argon. When a clear solution was obtained, the flask was cooled down to 0 °C using an ice bath and kept for another 15 min. Then methylmagnesium bromide solution in THF (20 mL) was added to the system dropwise with vigorous stirring. The ice bath was removed after complete addition of Grignard reagent. The reaction solution was further stirred for 24 h, resulting in the formation of a dark brown solution and then treated with 0.5 M HCl in methanol (30 mL) to prevent agglomeration (a red solution was formed). Next, another 0.5 M HCl in aqueous solution (50 mL) was added and the solution was stirred at 60 °C for 4 h. After THF was evaporated, the final product was filtered off and washed with excess water until the filtrate became pH neutral. K-PIM-1 was further treated by stirring in methanol overnight and dried at 120 °C for 24 h, showing a yellow powder (0.42 g, 84.9% yield).

OH-PIM-1 synthesis started by placing lithium chloride (0.21 g, 5.0 mmol) into a dry three-neck round-bottom flask. The elimination of moisture trapped in the system was done twice by flame drying, high vacuum, and argon gas filling. After that, K-PIM-1 (0.49 g, 1.0 mmol) and anhydrous THF (20 mL) were added to the flask. Grignard reaction was carried out similarly as in PIM-1. The final mixture was stirred until the formation of a consistent slurry. Next, acidic workup was performed by respective addition of 0.5 M HCl in methanol (30 mL) and 0.5 M HCl in aqueous solution to adjust pH in the range of 4-5. The colloidal solution was stirred at 60 °C for 4 h. After THF solvent was evaporated, the final product was filtered off and washed with excess water until the filtrate became pH neutral. OH-PIM-1 was further treated by stirring in methanol overnight and dried at 120 °C for 24 h (0.42 g, 79.8% yield).

### Synthesis of K-PIM-1 and OH-PIM-1 via non-solvent method

Fine PIM-1 (LT) powder (2.07 g, 4.5 mmol) was placed in a dry 250 mL three-necked round bottom flask under an inert atmosphere of argon. The flask was cooled down to 0 °C using an ice bath for 15 min coupled with mild stirring. Then methylmagnesium bromide solution in diethyl ether (30 ml) was added dropwise over 10 min. The heterogeneous solution was allowed to stir at room temperature for 24 h (reddish-brown solution gradually turned green). For the acidic workup, the colloidal solution was placed in an ice bath for 15 min. Then 0.5 M HCl in methanol solution (85 mL) was slowly added to the solution, followed by 0.5 M HCl in an aqueous solution (100 mL). Additional stirring at 60 °C for 4 h was required for complete conversion. The sample was filtered off and washed with an excess amount of water following by methanol. The final product was stirred in methanol to remove all residues trapped in the pores and dried at 120 °C for 24 h to afford K-PIM-1, showing a yellow powder (2.12 g, 95.5% yield).

To synthesize OH-PIM-1, lithium chloride (0.95 g, 22.5 mmol) was first introduced into a three-neck round bottom flask. The moisture was eliminated by flame drying, and the air inside was removed by a vacuum pump and replaced by argon gas. This process was done twice. To acquire the OH-PIM-1 product, K-PIM-1 (2.23 g, 4.5 mmol) replaced PIM-1 as a starting material in the Grignard reaction described above. Then acidic workup was performed similarly by adding 0.5 M HCl in methanol (85 mL) and 0.5 M HCl in aqueous solution to adjust pH in the range of 4-5. The colloidal solution was stirred at 60 °C for 4 h. The solids were dried in an oven at 120 °C overnight, giving OH-PIM-1 as a milky yellow solid (2.17 g, 91.6% yield).

### The study of volatile reagent effect in non-solvent post-modifications

The same batch of PIM-1 was used to prepare methyl K-PIM-1 (MeMgBr), ethyl K-PIM-1 (EtMgBr), isobutyl K-PIM-1 (*i*-BuMgBr), tertbutyl K-PIM-1 (*t*-BuMgCl), and phenyl K-PIM-1 (PhMgBr). Fine PIM-1 powder (0.52 g, 1.125 mmol) was placed in a dry 250 mL three-necked round bottom flask under an inert atmosphere of argon. The flask was

cooled down to 0 °C using an ice bath for 15 min coupled with mild stirring. Then different Grignard reagents (30 mL of methylmagnesium bromide solution (3.0 M) in diethyl ether for the synthesis of methyl K-PIM-1 (MeMgBr), 30 mL of ethylmagnesium bromide solution (3.0 M) in diethyl ether for the synthesis of ethyl K-PIM-1 (EtMgBr), 30 mL of phenylmagnesium bromide solution (3.0 M) in diethyl ether for the synthesis of phenyl K-PIM-1 (PhMgBr), 45 mL of isobutylmagnesium bromide solution in (2.0 M) diethyl ether for the synthesis of isobutyl K-PIM-1 (*i*-BuMgBr), 45 mL of tert-butylmagnesium chloride (2.0 M) in diethyl ether for the synthesis of tertbutyl K-PIM-1 (*t*-BuMgCl) were added dropwise over 10 min. The heterogeneous solution was allowed to stir at room temperature for 24 h (reddish-brown solution). For the acidic workup, the colloidal solution was placed in ice bath for 15 min. Then 0.5 M HCl in methanolic solution (85 mL) was slowly added to the solution. Each batch was gradually added 0.5 M HCl in an aqueous solution to have a final pH in the range of 4-5. Additional stirring at 60 °C for 4 h was required for complete conversion. The solids were filtered off and washed with an excess amount of water following by methanol. The final product was stirred in methanol to remove all residues trapped in the pores and dried at 120 °C for 24 h to afford K-PIM-1 derivatives.

### Amine functionalization

In a dry three-neck round bottom flask equipped with a reflux condenser, K-PIM-1 (0.74 g, 1.5 mmol) was suspended in ethanol (15 mL). The mild acidic condition was induced by adding glacial acetic acid (0.45 mL) and heating up to 80 °C under argon atmosphere. After the temperature reached the setpoint, different amine solutions (0.76 mL of phenylhydrazine reagent for the synthesis of HZ-PIM-1, 1.9 mL of methylamine solution for the synthesis of MI-PIM-1, or 5.7 mL of polyethylenimine for the synthesis of PEI-PIM-1) were added to the solution depending on the targeted final product, and the reaction was refluxed for 48 h. Then the reactions were cooled down to room temperature. The final products were filtered and washed with ethanol (4 × 50 mL). Additionally, the samples were stirred in ethanol (100 mL) for 20 min before being filtered and dried in the vacuum oven at 105 °C for 12 h, giving HZ-PIM-1 as a strong orange powder (0.84 g, 84.0% yield), MI-PIM-1 as a dark yellow powder (0.70 g, 90.0% yield), and PEI-PIM-1 as a mustard-yellow powder (0.94 g).

### Characterization

Elemental analyses were performed on an elemental analyzer (CHN-O) Thermo Fisher Scientific (Flash smart), measuring the elemental composition of the materials in the powder form with an accuracy of 0.3%. Each reported elemental compositions were from at least an average of two different measurements. Fourier transform infrared (FT-IR) spectra of powder samples were recorded on a SHIMADZU IRTracer-100 spectrometer with an Attenuated Total Reflectance accessory (GladiATR 10) in the 400 – 4000 $cm^{-1}$. Average molecular weights of the parent polymer were measured by gel permeation chromatography (GPC). Analysis was performed in THF. Thermogravimetric Analysis (TGA) was carried out using a High-Resolution TGA (Netzsch). Polymer samples were heated from 25 °C to 900 °C at a rate of 10 °C/min. For each sample, the test was performed in both $N_2$ and $O_2$ atmospheres. The ICP-MS instrument of Agilent 7700x model was used for magnesium content analysis. XPS (The Kratos Axis Supra) analysis performed at KAIST Analysis Center forResearch Advancement (KARA) was used to compare N1s XPS spectra of PIM-1 and K-PIM-1. [1]H NMR spectra of PIM-1 were obtained with a Bruker AVANCE 400 MHz using $CDCl_3$ as a solvent. Solid-state [13]C CP-MAS NMR spectra were achieved on a 400 MHz Bruker Avance III spectrometer with the spinning speed of 12 kHz to 20 kHz. The adsorption isotherms ($N_2$ and $H_2$ at 77 K), gas uptakes ($CO_2$ and $N_2$ at 273 K, 298 K, and 323 K), and $CH_4$ uptake at 273 K were obtained with a Micromeritics 3Flex surface characterization analyzer, Micromeritics Inc. Samples were degassed

at 110 °C for at least 12 hours under vacuum before the measurements. $CO_2$ breakthrough experiments were carried out in a custom made system attached to a QGA analyzer from Hiden analytical, UnitedKingdom. Schematics are provided in the Supplementary Information.

**Calculation of isosteric heats of adsorption ($Q_{st}$ values) and estimation of gas adsorption selectivity by Ideal Adsorption Solution Theory (IAST)**

The isosteric heat of adsorption ($Q_{st}$ values) is a key parameter determining energy consumption for adsorbent regeneration. To calculate $Q_{st}$, Clausius−Clapeyron equation (Eq. 1) has been employed, where $Q_{st}$ is the isosteric heat of adsorption, $R$ is the universal gas constant, and $T$ is the temperature, $p$ is the pressure for a given amount adsorbed $n$.

$$Q_{st} = RT^2 \left( \frac{\partial lnp}{\partial T} \right)_n \tag{1}$$

To obtain all required parameters, the adsorption data were collected at 273 K, 298 K, and 323 K. The entire set of isotherm data was fitted with a dual-site Langmuir−Freundlich (DSLF) model (Eq. 2) in Wolfram Mathematica.

$$Q = \frac{q_1 \times b_1 \times p^{1/n_1}}{1 + b_1 \times p^{1/n_1}} + \frac{q_2 \times b_2 \times p^{1/n_2}}{1 + b_2 \times p^{1/n_2}} \tag{2}$$

where $Q$ is the absorbed amount of adsorbate, $q_i$ is the saturation capacity, $p$ is the pressure, $b_i$ is Langmuir parameter of affinity coefficients related to the site, and $n_i$ represents deviations from an ideal surface. Lastly, $Q_{st}$ values obtained from the calculation was plotted as a function of $CO_2$ loading (Fig. 4d).

To calculate IAST binary-gas adsorption selectivities, the pure component isotherm data of $CO_2$ was fitted by means of the dual-site Langmuir−Freundlich (DSLF) model (Eq. 2), whereas $N_2$ and $CH_4$ were fitted by using a single-site Langmuir−Freundlich (SSLF) model (Eq. 3). The required parameters were obtained by using Wolfram Mathematica. The adsorption selectivity (S) for binary mixtures is calculated from Eq. 4.

$$Q = \frac{q_1 \times b_1 \times p^{1/n_1}}{1 + b_1 \times p^{1/n_1}} \tag{3}$$

$$S = \frac{q_1/q_2}{p_1/p_2} \tag{4}$$

where $q_i$ is the molar loading of a gas component in the adsorbed phase and $p_i$ is the partial pressure. The ratio of gas mixtures is at 15:85 for the calculation of $CO_2$ over $N_2$ selectivity (273 K, 298 K, and 323 K) and at 50:50 for the calculation of $CO_2$ over $CH_4$ selectivity (273 K).

Further details on the experimental methods can be found in the Supplementary Information.

## Data availability
The data sets generated and/or analyzed in this study are included in this published article and its supplementary information file. Additional source data are available from the corresponding author (C.T.Y.) upon request.

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

## Acknowledgements
This research was made possible by grants awarded to C.T.Y. from the King Abdullah University of Science and Technology (KAUST) and the Nano-Material Technology Development Program through the National Research Foundation of Korea (NRF) funded by the Ministry of Science and ICT (NRF-2017M3A7B4042140 and NRF-2017M3A7B4042273). B.K. acknowledges National Research Foundation of Korea (NRF) Grant of the Korean Government (No. 2020R1A4A1018516). S.W. also thanks to PTT Global Chemical Public Company Limited, Thailand for the GC scholarship.

## Author contributions
S.W. conducted all experiments, performed data analysis, and contributed to the writing and editing of the manuscript. T.S.N. assisted in setting up experiments, data analysis, and editing of the manuscript. T.P.N.N., A. A., and M. E. setup and analyzed breakthrough data. W.L. verified the equation and calculation of isosteric heats of adsorption and gas adsorption selectivity. Y.H. performed ICP-MS measurements. B.K. and J.P. performed GPC measurements. M.A. contributed to the discussion of breakthrough data. C.T.Y. conceived and supervised the project, and wrote the manuscript with input from all authors.

## Competing interests
The authors declare no competing interests.
