## [Peer Review File · Nature Communications]

Non-solvent post-modifications with volatile reagents for remarkably porous ketone functionalized polymers of intrinsic microporosityReviewers' Comments:

Reviewer #1:

Remarks to the Author:

In this manuscript, Yavuz and co-authors report a new modification strategy for PIM-1. PIM-1 is a well-known ladder polymer with permanent porosity. As the authors mentioned, it has been modified a number of times, including their own amidoxime version. In this work, they introduce 4 new functionalities which depend on making ketone functionality first. Interestingly, ketone or aldehyde functionalities on PIM-1 architecture are absent from the literature and the authors are achieving them with a simple modification. They show the versatility of ketone functionality by modifying with amines. The Schiff base is a common method of derivatization in porous materials but their approach of using non-solvents in a new angle. Since PIMs are soluble, it seems like it shows a dramatic difference. The second breakthrough they mention is the use of volatile reagents in post-modification. The methyl to phenyl reagent comparison indicates there is a lot happening and a potentially very useful concept is there. I believe this is more important and more applicable to the general porous materials field. The authors need to emphasize volatile reagent related findings more clearly and support with new data. Below are my requests for minor revision before publication:

1. The volatile reagent-based modification is very important and more reagents should be tested. How about a tert-butyl reagent? A bulky substrate (still volatile) may be able to get even more porosity?
2. BET is sensitive to the pressure ranges that are selected for the calculation. The authors need to report BET ranges for at least the top performing materials. Rouquerol plots would be even better. Also, they need to disclose their calculations in the supplementary.
3. BET depends a lot on the weight of the samples. Authors need to disclose the parameters they used. They're also mentioning that they used Wolfram Mathematica in the details of the Qst calculation (methods section in the main text). They need to elaborate further on these in the SI.
4. It might be helpful to display the mechanism for the ketone formation as a supplementary figure. It isn't that obvious to general readers.
5. FTIR (Figure 3a) is hard to read – use larger fonts and a longer column to separate/stack the plots
6. K-PIM-1 BET in Figure 3b and d should be identical? Why are they different?
7. Qst for PEI-PIM-1 shows expansion (binding energy increase over loading). There should be some discussion around this.
8. Figure 1 could use some professional recoloring/design.
9. Figure 2 doesn't need a title in the figure ("post-modification") and some text could be moved to the caption.

Reviewer #2:

Remarks to the Author:

The PIM-1 is a versatile polymer that has many advantages. The materials have robust stability due to the presence of rigid heterocyclic rings, a high BET surface area among the PIM, and great solubility which makes PIM-1 have good processability. The strategy that functionalizes the remaining nitrile group using tetrafluoroterephthalonitrile is quite common (Xinyu Guan et al. Nat. Chem., 2019, 11, 587; Zhongyue Li et al. Chem. Eng. J., 2022, 434, 134623) but in this paper, researchers successfully functionalized the ketones, alcohols, imines, and hydrazones for the first time. Moreover, I believe that this post-synthetic modification of the nitrile group with Grignard reagent has paved the way to functionalize various functional groups or introduce desired moieties. Moreover, in the previous research in respect to the PIM field, the modified PIM-1s were barely porous with tens of BET surface area after modification. However, in this research, the authors used heterogeneous modification and it showed a superior conversion ratio and porosity was more retained compared to the conventional homogeneous reaction method. Among the modified samples, the PEI-PIM exhibited dramatic CO₂ uptake enhancement compared to pristine PIM-1. Although the CO₂ capacity is low compared to other state-of-art CO₂ adsorbents, I believe that it is enough to verify the effect of functionalized group and

PIM-1 derivatives has other merits such as processability and stability. Overall, the paper could be considered after major improvements. Additional explanation and characterizations are required as follows.

1) The authors synthesized K-PIM-1 and OH-PIM-1 from the reaction between methyl magnesium bromide (in diethyl ether solution) and solid PIM-1 and named it "non-solvent chemistry" But author used solvent which did not dissolve polymer(PIM-1) and it is just a heterogeneous reaction (solid-liquid) which is commonly conducted in post-synthetic modification of insoluble porous materials such as MOFs, COFs, and POPs. So the term "non-solvent" is wrong and must be corrected throughout the paper to avoid the misunderstanding of readers.

2) From the XPS, elemental analysis, and N₂ isotherms the K-PIM-1 and OH-PIM-1 synthesized with ether solvents showed higher porosity compared to samples synthesized with homogeneous reaction even though the degree of conversion is higher. And the explanation of this counter-intuitive result was not enough. The authors only stated that "To attempt to improve conversions, we decided to focus on how we carry out the modifications. Traditional reactions usually involve the complete mixing of reagents in a solvent for uniform dissemination of reactive components (Fig. 1). However, in such homogeneous mixtures, the necessary molecular collisions are truly hindered because of the dominant presence of the solvents." on page 4. Generally, collision in the heterogeneous reaction would be more hindered compared to a homogeneous reaction. This phenomenon could be a solvent effect tentatively due to the high surface tension of THF (Dongyang Zhu et al. ACS Appl. Mater. Interfaces 2020, 12, 33121) or it could be ascribed to the difference of concentration in reaction (non-solvent method: 2.07g PIM-1 + CH₃MgBr solution 30 mL vs solvent method: 0.46 g PIM-1 + THF 20 mL +CH₃MgBr 20 mL). In summary, additional descriptions or control tests were required to explain the superiority of the "non-solvent method" (heterogeneous reaction).

3) In the heats of adsorption calculation section, the dual-site Langmuir-Freundlich (DSLFL) model fitting of isotherms is absent. It should be presented and the goodness of the fitting of both logarithmic-scale and linear-scale should be confirmed according to literature (Nuhnen A. et al., Dalton Trans., 2020, 49, 10295)

4) The figure of "2. Proof of concept "in the Si page 5 is not presented.

5) The authors stated that "The ideal adsorption solution theory (IAST) method was employed as it is the most reliable prediction for the actual gas separations." Indeed, IAST is a good method but to confirm the practical gas separation performance, breakthrough experiments with the commensurate gas mixture are required. In addition, other PEI functionalized CO₂ adsorbents often suffer from sluggish adsorption kinetics due to occlusion of pore and PEI-PIM-1 also exhibited low BET surface area. (49 m²/g) Therefore, the CO₂ breakthrough experiments were recommended to confirm the CO₂ separation capacity of PEI-PIM-1.

6) On page 14, the author stated that "Considering the CO₂ uptake at a low partial pressure (0.15 bar at 298 K) of any PIM base that has been reported to date, the uptake capacity of PEI-PIM-1 is one of the highest, and most certainly the most robust with the chemically tethered PEI 8, 14, 17, 18, 33, 40, 41" It would be better to tabulate the CO₂ capacity of PEI-PIM-1 and other PEI-tethered materials for the comparison.

7) The authors confirmed the post-synthetic modification mainly with characteristic FT-IR peaks and calculate the degree of conversion based on FT-IR peak ratios or elemental analysis which is not convincing. The NMR (carbon and nitrogen) and XPS data of each sample would be helpful to confirm the characteristic peak of introduced functional groups and further support the degree of conversion calculations.

8) On page 8, the authors stated that "The reaction between K-PIM-1 and phenylhydrazine yielded a

distinct orange solution (HZ-PIM-1), exhibiting visual proof of the presence of the carbonyl group (Supplementary Fig. 3).” As the rule of thumbs, color change often implied something changed but I think it is inappropriate to include it in the scientific manuscript. If the authors think visual change proves the modifications, UV-vis spectrum analysis is recommended.

Reviewer #3:

Remarks to the Author:

Processable porous materials with tunable functionality are of importance for applications in molecular separations, sensors, etc. This manuscript introduces an exciting new approach for chemically modifying the polymer of intrinsic microporosity PIM-1 that can be conducted with a non-solvent and could potentially be applied, for example, in the post-treatment of membranes. I believe this will generate a great deal of interest in the polymer, porous materials and membrane communities. However, some minor corrections are required:

Line 55 “Bekir et al.” should be “Satilmis et al.” (Bekir is his given name).

Line 79: “called as a” would be better “called a”

Line 146: “occurred to be effortless” might be better “occurred readily” or something similar

Line 151: “These evidences” would be better “This evidence” (“evidence” is a body of information, “evidences” is a verb meaning to indicate clearly)

Line 166: “the pristine PIM-1 is incapable of reacting with amines” Although this is probably true under the conditions utilized, it might be appropriate to mention that there is the possibility of PIM-1 reacting with PEI on heat treatment, as suggested by Putintseva et al., Polymer Science, Series B, 2019, Vol. 61, No. 6, pp. 795–805.

Line 185: “Characterizations” does not need to be plural (although one could talk about “characterization methods”)

Line 186: “BET Surface area isotherms using N₂ as a probe gas” This reviewer would prefer to refer to the isotherms as “N₂ adsorption/desorption isotherms” and restrict the term “BET surface area” to the apparent surface areas calculated by applying the BET equation to a portion of the isotherm.

Line 188: “Comparison of surface areas between” This reviewer would say “Comparison of N₂ adsorption/desorption isotherms and BET surface areas for”

Line 201: “effect of the volatility for modifying agents” Is the difference in BET surface area really due to the volatility of the reagent, and not the bulkiness of the side group that occupies free volume in the product?

Line 220-221: “A hysteresis loop was observed in all nitrogen isotherms, suggesting the presence of mesopores” The type of hysteresis observed here, which extends to low relative pressures, does NOT suggest mesopores. Mesoporosity is associated with a type of hysteresis that closes at a relative pressure of 0.4 or above. Low pressure hysteresis is characteristic of PIMs and may be attributed either to swelling of the polymer by the adsorbate or to a complex micropore structure.

Line 223: “displayed” might be better “is demonstrated by”

Line 229-230: “mesopore cavity” There is no evidence of mesopores, see above comment on line 220-221. Caution should be exercised in interpreting NLDFT pore size distributions for these types of

isotherms. Swelling by adsorbate may be misinterpreted as mesoporosity.

Line 233: "increase the micropores" This reviewer is not convinced the evidence is strong enough to make this claim.

Line 240: Table 1 "PSD" is not defined. If this is meant to indicate a pore size, this reviewer is not convinced (see comments above).

Line 334: "would be extended" should perhaps be "could be extended"

Responses to Reviewer Comments

Manuscript number: NCOMMS-22-03122

Reviewer 1

Reviewer recommendation term: Minor Revision

In this manuscript, Yavuz and co-authors report a new modification strategy for PIM-1. PIM-1 is a well-known ladder polymer with permanent porosity. As the authors mentioned, it has been modified a number of times, including their own amidoxime version. In this work, they introduce 4 new functionalities which depend on making ketone functionality first. Interestingly, ketone or aldehyde functionalities on PIM-1 architecture are absent from the literature and the authors are achieving them with a simple modification. They show the versatility of ketone functionality by modifying with amines. The Schiff base is a common method of derivatization in porous materials but their approach of using non-solvents in a new angle. Since PIMs are soluble, it seems like it shows a dramatic difference. The second breakthrough they mention is the use of volatile reagents in post-modification. The methyl to phenyl reagent comparison indicates there is a lot happening and a potentially very useful concept is there. I believe this is more important and more applicable to the general porous materials field. The authors need to emphasize volatile reagent related findings more clearly and support with new data. Below are my requests for minor revision before publication:

We are truly grateful to the reviewer for his/her comments, for taking the time to evaluate our work, and for acknowledging that this is an interesting topic. New data has been supplemented following the reviewer's comments as follows:

1.1) The volatile reagent-based modification is very important and more reagents should be tested. How about a tert-butyl reagent? A bulky substrate (still volatile) may be able to get even more porosity?

We appreciate the insightful comment and advice from the reviewer. We agree that additional reagents should be tested. As suggested by the reviewer, we have now chosen to test a tert-butyl reagent – tert-butyl magnesium chloride (*t*-BuMgCl – bromide salt is not commercially available). In addition, we have also tested two more commercially available Grignard reagents: ethyl magnesium bromide (EtMgBr), isobutyl magnesium bromide (*i*-BuMgBr). The results are now combined in a revised Figure 3:

Fig. 3 | Characterization and volatile reagent effects in the non-solvent post-modification. **a** FT-IR spectra and **b** N_2 adsorption/desorption isotherms measured at 77 K. **c** Comparison of N_2 adsorption/desorption isotherms and BET surface areas for post-modifications of PIM-1 by volatile and non-volatile reagents via the non-solvent method. and **d** Schematic illustration of the post-synthetic modification by using different modifying substances.

In summary, based on the same series of Grignard reagents (bromide salts), the modifying group was orderly varied by the volatility of the decomposition product (conjugate hydrocarbon): methyl>ethyl>isobutyl>phenyl. A highly volatile reagent yields the highest surface area. In the case of tert-butylmagnesium chloride, we saw a battle between bulkiness and volatility, leading to an in between porosity, when compared to MeMgBr and PhMgBr. The main text was modified accordingly as follows:

“To investigate the effect of volatility in K-PIM-1 formation, we used the same batch of PIM-1 to treat with methylmagnesium bromide (MeMgBr), ethylmagnesium bromide (EtMgBr),

isobutylmagnesium bromide (*i*-BuMgBr) and phenylmagnesium bromide (PhMgBr) in identical non-solvent conditions (Fig. 3d).”

“As expected, PIM-1 modified by volatile reagents: MeMgBr, EtMgBr, and *i*-BuMgBr showed high surface areas (S_{BET}: 701 m²/g, 573 m²/g, and 346 m²/g, respectively). In comparison, phenyl ketone product from the non-volatile PhMgBr reagent came out to have no surface area. This indicates that the surface area significantly depends on the volatility of the modifying group. Highly volatile reagents yield higher surface areas. Further investigation was thoroughly performed by using tert-butylmagnesium chloride (*t*-BuMgCl), a bulky volatile reagent. Surprisingly, the bulky volatile reagent of *t*-BuMgCl could maintain the surface area (425 m²/g), even though bulky groups usually hinder the porosity of porous polymers by pore filling. Since the full conversion to K-PIM-1 by *t*-BuMgCl was not achieved due to steric hindrance, half conversion of a non-volatile K-PIM-1 (PhMgBr) was additionally performed to make a fair comparison and examine the porosity retention of the final product. It was clearly seen that K-PIM-1 (PhMgBr) at half conversion was also virtually non-porous (S_{BET}: 9 m²/g) (Fig. 3c and Supplementary Fig. 15-16).”

We thank the reviewer for this contribution.

1.2) BET is sensitive to the pressure ranges that are selected for the calculation. The authors need to report BET ranges for at least the top performing materials. Rouquerol plots would be even better. Also, they need to disclose their calculations in the supplementary.

We agree with the reviewer’s comment. We have now performed additional BET analysis at ultra-low pressures to acquire better gas adsorption isotherms and to use correct range determination through *Rouquerol* plots. Although there is no significant change in surface areas, we updated the data throughout the manuscript accordingly. The calculation of specific surface area can be found in supplementary information section 4. The Rouquerol and BET plots that include the selected points and pressure ranges, were reported in detail at the supplementary information, Supplementary Figures 12-22. We’re now including the calculation methods and the related figures for all structures in the SI, but showing only one example here to eliminate the clutter:

“4. Calculation of Surface Areas^{5,6}

To calculate the surface areas from N₂ adsorption isotherms at 77 K, we applied following criteria:

- 1) The linear fit should span at least 5 points.
- 2) The R² should be greater or equal to 0.995.
- 3) Over the entire fitting range Q(1-P/P₀) must continuously increase with P/P₀
- 4) The value of C intercept obtained by linear regression must be positive in the plot

of $1/[Q(P_0/P-1)]$ against P/P_0

The amount of gas molecules adsorbed in the initial monolayer is

$$Q_m = \frac{1}{\text{Slope} + \text{Intercept}} \quad (1)$$

The specific surface area was calculated as follows:

$$SA_{BET} = Q_m \left(\frac{\text{cm}^3}{\text{g}} \right) \times \frac{1 \text{ (mol)}}{22400 \text{ (cm}^3\text{)}} \times 16.2 \text{ (\AA}^2\text{)} \times N_A \text{ (mol}^{-1}\text{)} \times 10^{-20} \left(\frac{\text{m}^2}{\text{\AA}^2} \right) \quad (2)$$

Where N_A is Avogadro's constant, and 16.2 \AA^2 is the cross-sectional area of a N_2 molecule."

Supplementary Figure 12: a Calculated Rouquerol plot for K-PIM-1 along with the pressure ranges used for BET surface area calculations. **b** BET plot of K-PIM-1 obtained from N_2 adsorption isotherm at 77 K.

New references in SI:

- Osterrieth, J.W.M., et al. How Reproducible are Surface Areas Calculated from the BET Equation? *Adv. Mater.* **34**, (2022).
- Rouquerol, J., Llewellyn, P., Rouquerol, F. Is the BET equation applicable to microporous adsorbents? *Stud. Surf. Sci. Catal.* **160**, 49-56 (2006).

The following text was also added to the main text to reflect these discussions:

“The surface areas of all PIM-1s and derivatives were determined using BET theory on N₂ adsorption-desorption isotherms at 77 K (Fig. 3b and Supplementary Fig. 17-22). The calculations included determination of the ideal pressure range through Rouquerol plots (detailed calculations can be found in section 4 of the supplementary information). We have used best fitting values as it is well known that BET areas could deviated significantly just by simple data point exclusions^{36, 37}.”

New references in the manuscript:

36. Osterrieth, J.W.M., et al. How Reproducible are Surface Areas Calculated from the BET Equation? *Adv. Mater.* **34**, (2022).

37. Rouquerol, J., Llewellyn, P., Rouquerol, F. Is the BET equation applicable to microporous adsorbents? *Stud. Surf. Sci. Catal.* **160**, 49-56 (2006).

Once again, we thank the reviewer for the contribution.

1.3) BET depends a lot on the weight of the samples. Authors need to disclose the parameters they used. They're also mentioning that they used Wolfram Mathematica in the details of the Q_{st} calculation (methods section in the main text). They need to elaborate further on these in the SI.

We appreciate the reviewer's careful attention. We are now reporting sample weights used for BET analysis and all parameters for calculation in Supplementary Figures 12-22 (see above Supplementary Figure 12 text inset for example). For Q_{st} calculation using Wolfram Mathematica, all required parameters are disclosed in the supplementary information. In addition, we also include the fitting data by using OriginPro as an alternative approach. The updated results can be found in Supplementary Figure 26-33 (only one example shown here as reference) and Supplementary Table 11-14. We thank the reviewer for helping us be transparent in our calculations and models. The following text was revised for better reflect this discussion:

“During this calculation, we also demonstrated an alternative approach by performing Dual-Site Langmuir-Freundlich (DSLFF) model fitting for CO₂ adsorption isotherms at 273 K, 298 K, and 323 K via logarithmic-scale plots using OriginPro. The fitted parameters were also used to calculate Q_{st} value (Supplementary Fig. 32-33, and Supplementary Table 13-14). The experimental data sets were fitted well for both Wolfram Mathematica and OriginPro. The Q_{st} values obtained from both sources revealed the same tendency.”

Supplementary Figure 26: CO₂ adsorption isotherms of PIM-1 at a 273 K, b 298 K, c 323 K fit to the Dual-Site Langmuir-Freundlich (DSLFL) model by Wolfram Mathematica.

Supplementary Figure 32: Logarithmic-scale plots for Dual-Site Langmuir-Freundlich (DSLFL) model fitting for CO₂ adsorption isotherms at 273 K, 298 K, and 323 K by OriginPro on **a** PIM-1, **b** K-PIM-1, **c** OH-PIM-1, **d** MI-PIM-1, **e** HZ-PIM-1, and **f** PEI-PIM-1 (An alternative approach).

1.4) It might be helpful to display the mechanism for the ketone formation as a supplementary figure. It isn't that obvious to general readers.

We thank the reviewer for his/her suggestion. The mechanism for the ketone formation is shown below and added in the supplementary information (Supplementary Figure 4).

Supplementary Figure 4: The mechanism for the ketone formation.

1.5) FTIR (Figure 3a) is hard to read – use larger fonts and a longer column to separate/stack the plots.

We thank the reviewer for his/her kind suggestion. We revised Figure 3a and are attaching it here for reference:

Fig. 3a. FT-IR spectra.

1.6) K-PIM-1 BET in Figure 3b and d should be identical? Why are they different?

We appreciate the reviewer for pointing out the surface area value differences for different batches of K-PIM-1. During our studies, we have synthesized K-PIM-1 in a number of batches to ensure reproducibility. We are now including the variations in batches in a Supplementary Table 2. Consequently, the difference was because the K-PIM-1 in Figure 3b and 3d were from different synthesis batches. For Figure 3d, we used the same batch of PIM-1 to treat with both CH_3MgBr and PhMgBr in identical non-solvent conditions to avoid any deviation, while K-PIM-1 BET reported in Figure 3b is the highest surface area of K-PIM-1 among the 5 batches of K-PIM-1 synthesis. To avoid further confusion, we have now reported the same value in both figures by using the surface area that obtained from the latest experiment ($701 \text{ m}^2/\text{g}$ instead

of either 715 or 705 m²/g). We have updated the Figure 3 (see above) accordingly. We thank the reviewer for preventing an unnecessary confusion.

1.7) Q_{st} for PEI-PIM-1 shows expansion (binding energy increase over loading). There should be some discussion around this.

We thank the reviewer for highlighting this point. We kindly note that the tendency of Q_{st} for CO₂ adsorption can be explained by the combination between the interactions of adsorbent-adsorbate and adsorbate-adsorbate. Due to high favorability of PEI-PIM-1 towards CO₂, upon increasing the CO₂ loading, CO₂ molecules largely occupied within its structure, enhancing the interaction between CO₂ molecules. At higher coverage, the CO₂ molecules can force expansion of the flexible PEI chains, strengthening their binding by additional access to additional nitrogens and the ensuing complexation (e.g. wrapping mechanism that we previously reported in *Phys. Chem. Chem. Phys.*, 2016,18, 14177-14181). As a consequence, the Q_{st} for PEI-PIM-1 shows expansion as a result of larger heat release.

We add the following note to the manuscript in the section of gas uptake studies to discuss this phenomenon coupled with the references as listed below:

“As predicted, covalent immobilization of PEI through ketone functionality of K-PIM-1 could tune binding energy into an ideal range ($Q_{st,max} = 44.5$ kJ/mol). PEI-PIM-1 also showed expansion of Q_{st} at higher CO₂ loading. This is due to high favorability of PEI-PIM-1 towards CO₂. Upon increasing the CO₂ loading, CO₂ molecules largely occupied within its structure, enhancing the interaction between adsorbent-adsorbate and adsorbate-adsorbate. At higher coverage, the CO₂ molecules can force expansion of the flexible PEI chains, strengthening their binding by additional access to more nitrogens and the ensuing complexation⁶³. Ultimately, the interaction between CO₂ molecules and the polyamines can be continuously strengthened to increase in Q_{st} ^{64, 65}.”

New references in the manuscript:

63. Thirion, D., et al. Observation of the wrapping mechanism in amine carbon dioxide molecular interactions on heterogeneous sorbents. *Phys. Chem. Chem. Phys.* **18**, 14177-14181 (2016).
64. Simmons, J.M., Wu, H., Zhou, W., Yildirim, T. Carbon capture in metal-organic frameworks-a comparative study. *Energy Environ. Sci.* **4**, 2177-2185 (2011).
65. Qazvini, O.T., Babarao, R., Telfer, S.G. Selective capture of carbon dioxide from hydrocarbons using a metal-organic framework. *Nat. Commun.* **12**, (2021).

1.8) Figure 1 could use some professional recoloring/design.

We appreciate the reviewer for this valuable suggestion. We have revised Figure 1 to be clearer.

Fig. 1 | Conceptual representation of the synthesis options for a successful modification of soluble porous materials without losing porous properties: conventional synthetic method (left), solventless mechanochemistry (right), and our proposed non-solvent approach (middle).

1.9) Figure 2 doesn't need a title in the figure (“post-modification”) and some text could be moved to the caption.

We thank the reviewer for pointing this out and the suggestion. We have changed Figure 2 accordingly by omitting the title and removed the unnecessary text.

Fig. 2 | The quantitative post-modification of PIM-1 through conventional and non-

solvent methods with a volatile Grignard reagent, CH₃MgBr. K-PIM-1 was further reacted by the same reagent to make OH-PIM-1 with remarkably high retention of porosity, considering the two-step reaction. **Solvent:** (i) CH₃MgBr in THF, THF solvent at 0 °C → RT for 24 h (ii) 0.5 M HCl in MeOH, H₂O at 60 °C for 4 h. **Non-solvent:** (i) CH₃MgBr in Et₂O at 0 °C → RT for 24 h (ii) 0.5 M HCl in MeOH, H₂O at 60 °C for 4 h. K-PIM-1 was further functionalized by hydrazine and PEI to show versatility in the reactive portfolio and to create CO₂ adsorbents.

Reviewer 2

Reviewer recommendation term: Major Revision

The PIM-1 is a versatile polymer that has many advantages. The materials have robust stability due to the presence of rigid heterocyclic rings, a high BET surface area among the PIM, and great solubility which makes PIM-1 have good processability. The strategy that functionalizes the remaining nitrile group using tetrafluoroterephthalonitrile is quite common (Xinyu Guan et al. Nat. Chem., 2019, 11, 587; Zhongyue Li et al. Chem. Eng. J., 2022, 434, 134623) but in this paper, researchers successfully functionalized the ketones, alcohols, imines, and hydrazones for the first time. Moreover, I believe that this post-synthetic modification of the nitrile group with Grignard reagent has paved the way to functionalize various functional groups or introduce desired moieties. Moreover, in the previous research in respect to the PIM field, the modified PIM-1s were barely porous with tens of BET surface area after modification. However, in this research, the authors used heterogeneous modification and it showed a superior conversion ratio and porosity was more retained compared to the conventional homogeneous reaction method. Among the modified samples, the PEI-PIM exhibited dramatic CO₂ uptake enhancement compared to pristine PIM-1. Although the CO₂ capacity is low compared to other state-of-art CO₂ adsorbents, I believe that it is enough to verify the effect of functionalized group and PIM-1 derivatives has other merits such as processability and stability. Overall, the paper could be considered after major improvements. Additional explanation and characterizations are required as follows.

We truly appreciate the reviewer for his/her time and for careful evaluation as well as giving constructive comments, which has helped greatly to improve our manuscript.

2.1) The authors synthesized K-PIM-1 and OH-PIM-1 from the reaction between methyl magnesium bromide (in diethyl ether solution) and solid PIM-1 and named it “non-solvent chemistry” But author used solvent which did not dissolve polymer(PIM-1) and it is just a heterogeneous reaction (solid-liquid) which is commonly conducted in post-synthetic modification of insoluble porous materials such as MOFs, COFs, and POPs. So the term “non-solvent” is wrong and must be corrected throughout the paper to avoid the misunderstanding of readers.

We appreciate the reviewer’s suggestion and we understand that the definition of heterogenous

reaction can also apply for our proposed post-synthetic modification. However, the term of heterogenous reaction is quite broad. For a heterogenous reaction, the reactants are in different phases from each other, which can be between solid-gas, solid-liquid, two immiscible liquids and so on. To be more specific and correlate well with the goals of our work, we chose to use the term of “non-solvent chemistry” for our chemical modification strategy as the solid is intentionally made insoluble by a solvent choice. We are not the first to use it as this term can commonly be found in the literature (see for example, M. Talha Gokmen, Filip E. Du Prez, Porous polymer particles—A comprehensive guide to synthesis, characterization, functionalization and applications, Progress in Polymer Science, (2012), 37, 365-405 and Byeongdu Lee, Kenneth Littrell, Yuchen Sha, Elena V. Shevchenko, Revealing the Effects of the Non-solvent on the Ligand Shell of Nanoparticles and Their Crystallization, J. Am. Chem. Soc. 2019, 141, 42, 16651–16662). We hope that the reviewer agrees with our terminology choice and allow us to use in this work.

2.2) From the XPS, elemental analysis, and N₂ isotherms the K-PIM-1 and OH-PIM-1 synthesized with ether solvents showed higher porosity compared to samples synthesized with homogeneous reaction even though the degree of conversion is higher. And the explanation of this counter-intuitive result was not enough. The authors only stated that "To attempt to improve conversions, we decided to focus on how we carry out the modifications. Traditional reactions usually involve the complete mixing of reagents in a solvent for uniform dissemination of reactive components (Fig. 1). However, in such homogeneous mixtures, the necessary molecular collisions are truly hindered because of the dominant presence of the solvents." on page 4. Generally, collision in the heterogeneous reaction would be more hindered compared to a homogeneous reaction. This phenomenon could be a solvent effect tentatively due to the high surface tension of THF (Dongyang Zhu et al. ACS Appl. Mater. Interfaces 2020, 12, 33121) or it could be ascribed to the difference of concentration in reaction (non-solvent method: 2.07g PIM-1 + CH₃MgBr solution 30 mL vs solvent method: 0.46 g PIM-1 + THF 20 mL +CH₃MgBr 20 mL). In summary, additional descriptions or control tests were required to explain the superiority of the "non-solvent method" (heterogeneous reaction).

We appreciate the reviewer for his/her insightful comment and excellent advice. We agree with the reviewer that the unusual behavior we observed could have more than one feasible mechanism at play. We wanted to simplify by highlighting what we thought would be most effective but we see that it would be more fitting to give a breadth of factors rather than singling out one. We believe the following are the chief factors for our observations (there are potentially more possibilities but these would constitute a significant contribution):

- 1) Variation in surface tension of the competing solvents: As indicated by the reviewer, the surface tension differences could create different buoyancy in the solids dispersed in a solvent system. These forces would lead to improved delivery of the solutes, in this case the modifying agents.
- 2) Concentration difference and gradient: In Grignard modifications where volatile agents are used, excess reactant additions are common. This would inevitably create

concentration differences when solvent(s) are included or not. Also, the concentration gradients are unavoidable since there is more than one phase changing behavior is happening (e.g. gas evolution).

- 3) Kinetics and enhanced collision probabilities in a non-solvent system: This is what we think is the main reason for enhanced modifications since it samples all the other effects into a single event. The kinetics of collision is enhanced by the concentration effects, solvent surface tension and buoyancy. In a perfectly homogeneous system, the mixture is devoid of such driving forces and forces the reaction into a complete collision probability game.
- 4) Coordination or the lack thereof with solvent-solvate interactions: As a follow up on the concentration interplay, the coordination or screening by solvent-solvate interactions will hinder the necessary contact between reagents, preventing (or rather slowing) the chemical modifications.
- 5) Volatile reagent evolution driven equilibrium dynamics: The by-product in Grignard reactions is usually a hydrocarbon and, in our experiments, we intentionally chose the volatile ones. This was to move the equilibrium on deprotonation steps forward. The same gas evolution could also impact on the non-solvent mediated heterogeneous interactions, even more so than the homogeneous solutions. This is because the already imbalanced equilibrium is further shot forward in the favor of the modified products.

Based on this fruitful discussion, we're updating the corresponding text as follows:

"However, in such homogeneous mixtures of soluble porous organic polymers, the necessary molecular collisions are truly hindered because of the dominant presence of the solvents. In addition, fully solvated substances would not be affected by the positive driving force stemming from additional buoyancy from surface tension variations²⁷. The concentration gradients will also be lost due to effective mixing, removing the possibility of densely localized reactive media. Volatile reagents would also be better stabilized in a compatible solvent mixture, withdrawing the chances of a fast evolution driven reaction kinetics. The solubility of PIMs that derived from the interaction between polymer chain of PIMs and solvent could also become a barrier, limiting the reaction between PIMs and modifying reagent."

The reference provided by the reviewer was also included as follows:

27. Zhu, D., Verduzco, R. Ultralow Surface Tension Solvents Enable Facile COF Activation with Reduced Pore Collapse. *ACS Appl. Mater. Interfaces*. **12**, 33121-33127 (2020).

We thank the reviewer for this contribution.

2.3) In the heats of adsorption calculation section, the dual-site Langmuir-Freundlich (DSLFF) model fitting of isotherms is absent. It should be presented and the goodness of the fitting of both logarithmic-scale and linear-scale should be confirmed according to literature (Nuhnen A. et al., Dalton Trans., 2020, 49, 10295)

We appreciate the reviewer's suggestion and would be happy to provide the fitting data of isotherms. We have now extended our data to show the DSLF fitting of isotherms by Wolfram Mathematica. We also presented the logarithmic-scale plots and linear- scale by OriginPro according to literature (Nuhnen A. et al., Dalton Trans., 2020, 49, 10295), recommended by the reviewer. All isotherms are fitted well with both programs. However, the parameters obtained from both sources are different. Regarding the difference, we realize and thus extend to calculate Q_{st} by using parameters obtained from fitting by OriginPro. The Q_{st} values from both approaches showed relatively similar tendency. In order to avoid confusion for readers, we mention the calculation by OriginPro as an alternative method. All the additional data can be found in supplementary information. The updated results can be found in Supplementary Figure 26-33 and Supplementary Table 11-14.

Please find below the Supplementary Figure 27 and Supplementary Table 11 (K-PIM-1 part) as examples for the newly included data in SI:

Supplementary Figure 27: CO₂ adsorption isotherms of K-PIM-1 at **a** 273 K, **b** 298 K, **c** 323 K fit to the Dual-Site Langmuir-Freundlich (DSLF) model by Wolfram Mathematica

Supplementary Table 11: Comparison of Q_{st} values with three temperatures with Wolfram Mathematica for K-PIM-1.

K-PIM-1				
CO ₂ loading (mmol/g)	P@273	P@298	P@323	Q _{st} (kJ/mol)
0.05	0.008	0.027	0.065	30.13
0.1	0.017	0.054	0.132	29.99
0.15	0.026	0.082	0.201	29.84
0.2	0.036	0.112	0.273	29.69
0.25	0.046	0.142	0.347	29.55
0.3	0.057	0.173	0.424	29.41
0.35	0.068	0.206	0.503	29.27
0.4	0.080	0.240	0.585	29.13
0.45	0.093	0.274	0.671	29.00
0.5	0.106	0.311	0.759	28.87
0.55	0.120	0.348	0.851	28.75

We thank the reviewer for this suggestion.

2.4) The figure of “2. Proof of concept” in the Si page 5 is not presented.

We thank the reviewer for his/her kind reminder. We thought that the SI figure of “2. Proof of concept” on page no. 5 was displayed properly in the original submission. We apologize for this. It might be from downloading issues or pdf conversions. We have now made sure to double check that it shows up in the pdf forms and also attaching the figure in the section “2. Proof of concept” here again for the reviewer’s convenience:

Proof of concept control study figure in the supplementary information, section 2.

2.5) The authors stated that “The ideal adsorption solution theory (IAST) method was employed as it is the most reliable prediction for the actual gas separations.” Indeed, IAST is a good method but to confirm the practical gas separation performance, breakthrough experiments with the commensurate gas mixture are required. In addition, other PEI functionalized CO₂ adsorbents often suffer from sluggish adsorption kinetics due to occlusion

of pore and PEI-PIM-1 also exhibited low BET surface area. (49 m²/g) Therefore, the CO₂ breakthrough experiments were recommended to confirm the CO₂ separation capacity of PEI-PIM-1.

We appreciate the reviewer for his/her insightful comment. We agree that breakthrough experiments would add significant value in gas separation reports. We have now performed breakthrough experiments for both PEI-PIM-1 and the parent PIM-1. The details of CO₂ breakthrough experiments can be found in Supplementary section 5. The test results can be seen in the new Supplementary Figure 38:

5. CO₂ Breakthrough Experiment

5.1 Methodology

Well-ground polymer was filled into a stainless-steel column and activated at 120 °C for 2 hours in a tubular furnace (TF55035C-1, Thermo Scientific™) under 50 cm³/min flow of Helium gas (99.999%, Air Liquide) prior to breakthrough adsorption test. The test was carried out at 1 bar pressure and ambient temperature approximately 22 °C using 15%/85% CO₂/N₂ mixture (Gulf Cryo, Saudi Arabia) as adsorbates and Helium (99.999%, Air Liquide) as a carrier gas. The ratio of mixed gas and carrier gas was 1:1, and total gas flowrate passing through the column was 20 cm³/min. Outcoming gases were analyzed by a gas analysis system (QGA, Hiden analytical, United Kingdom).

Photograph and schematics of the breakthrough adsorption of CO₂/N₂ mixture on a fixed bed column.

5.2 Derivation of adsorption capacity

Adsorption capacity is calculated as the amount of uptake adsorbate per unit mass of the adsorbent.

$$q_t = \frac{m_{ad}}{m} \quad (1)$$

While,

$$m_{ad} = C_{ad}V \quad (2)$$

$$V = Ft \quad (3)$$

$$C_{ad} = C_o - C_t \quad (4)$$

Substitute (2), (3) and (4) into (1).

$$q_t = \frac{F}{m} (C_o - C_t)t \quad (5)$$

Then total adsorbate will be calculated by Eq. (6)

$$q = \frac{F}{m} \int_0^{t_s} (C_o - C_t) dt$$

$$\text{Or } q = \frac{FC_o}{m} \int_0^{t_s} \left(1 - \frac{C_t}{C_o}\right) dt \quad (6)$$

Where,

q : Adsorption capacity (mmol/g)

q_t : Adsorption capacity at time t (mmol/g)

m_{ad} : Mass of adsorbate (g)

m : Mass of adsorbent (g)

C_{ad} : Concentration of adsorbate taken by adsorbent (g)

V : Volume of adsorbate (L)

F : Flow rate of adsorbate (L/min)

t : adsorption time (min)

C_o : Initial concentration of adsorbate (mmol/L)

C_t : Concentration of adsorbate come out from fixed bed (mg/L)

t_s : Saturated time (min)

5.3 Calculation from experimental data:

$$F = 10 \text{ mL/min CO}_2 = 0.01 \text{ L/min}$$

$$C_o = \frac{1}{24.54 \text{ L}} \times 15\% = 6.112 \text{ mmol/L}$$

(At 1 bar, 22 °C, 1 mole of ideal gas is 24.54 L)

$m = 150 \text{ mg}$ of PIM-1 or PEI-PIM-1

$\int_0^{t_s} \left(1 - \frac{C_t}{C_o}\right) dt$ is the subtracted area between the plot $\frac{C}{C_o} = 1$ and $\frac{C}{C_o} = \frac{C_t}{C_o}$ versus time t

$$\text{For PIM-1, } \int_0^{t_s} \left(1 - \frac{C_t}{C_o}\right) dt = 4.863$$

Then adsorption capacity of PIM-1 is $q = 1.982 \text{ mmol/g}$

$$\text{For PEI-PIM-1 } \int_0^{t_s} \left(1 - \frac{C_t}{C_o}\right) dt = 7.358$$

Then adsorption capacity of PEI-PIM-1 is $q = 2.998 \text{ mmol/g}$

Supplementary Figure 38: CO₂ adsorption breakthrough curves on PIM-1 and PEI-PIM-1 at ambient temperature approximately 22 °C using a 15%/85% CO₂/N₂ mixture

The new results were mentioned in the main text as follows:

“To further appraise the performance of PEI-PIM-1 compared with PIM-1 in the actual adsorption-based separation process, CO₂ breakthrough experiments for PEI-PIM-1 and PIM-1 were tested at ambient temperature (approximately 22 °C) using a 15%/85% CO₂/N₂ mixture. As can be seen in Supplementary Fig. 38, PEI-PIM-1 had a substantially better separation performance compared to the parent PIM-1. The adsorption capacity of PEI-PIM-1 yielded at 2.998 mmol/g while PIM-1 was 1.982 mmol/g (see Supplementary Information section 5 for methods and calculations). While the uptake values are significantly higher than the pure gas isotherms, the percent increase in capacities remain similar. This is in line with the mixed gas observations in the literature¹⁵.”

We thank the reviewer for this contribution.

2.6) On page 14, the author stated that "Considering the CO₂ uptake at a low partial pressure (0.15 bar at 298 K) of any PIM base that has been reported to date, the uptake capacity of PEI-PIM-1 is one of the highest, and most certainly the most robust with the chemically tethered PEI 8, 14, 17, 18, 33, 40, 41” It would be better to tabulate the CO₂ capacity of PEI-PIM-1 and other PEI-tethered materials for the comparison.

We appreciate the reviewer for his/her recommendation. We prepared a list of comparisons and combined into a comparison graph for an insightful evaluation. The following Supplementary Figure 25 and Supplementary Table 10 are now added to the SI:

Supplementary Figure 25: Benchmarking CO₂ uptake at a low partial pressure versus Q_{st} values with other reported adsorbents.

Supplementary Table 10: Comparison of CO₂ capture performance of covalently tethered PEI.

Materials	CO ₂ adsorption			CO ₂ /N ₂ at 25 °C	Q _{st} (kJ/mol)	Reference
	Capacity (mmol/g)	Temperature (°C)	Pressure			
PEI-PIM-1	1.70 ^a	25	1.01 bar	316 ^h	44.5	This work
PP-AM-PEI fiber	5.91 ^b	25	-	-	-	Wu Q. et al. ¹⁵
PEI-MIL-101-100	5.00 ^c	25	1 bar	600 ⁱ	-	Lin Y. et al. ¹⁶
HG-PEI-1.98	4.13 ^d	25	1 atm	-	68 ^k	Liu F. et al. ¹⁷
PEI70@PGD-H	4.17 ^a	25	1 atm	73.3	48..4 ^l	Zhu J. et al. ¹⁸
Porous PS-PEI	3.46 ^a	40	1 bar	26.9 ^j	-	Liu Z. et al. ¹⁹
PEI50/AlSiO ₂ -TPI	2.51 ^e	50	-	-	-	Kolle J. M. et al. ²⁰
GMA-PEI (NUT-10)	2.09 ^a	0	1 bar	308	49	Mane S. et al. ²¹
PEI/SiO ₂	1.98 ^f	60	-	-	66.2 ^m	Min K. et al. ²²
EB-PEI/SiO ₂	1.62 ^f	60	-	-	80.5 ^m	Min K. et al. ²²
CNT-PEI	0.98 ^g	70	~1 atm	-	-	Zhou Z. et al. ²³
PEI grafted MCM-41(M1)	<1 ^a	30	1.01 bar	-	-	Kassab H. et al. ²⁴

^aAdsorption isotherm (Pure CO₂)

^bCO₂ breakthrough experiment (10%CO₂/N₂)

^cThe CO₂ adsorption kinetics of the probe gas CO₂ were also measured using volumetric technique by the apparatus from SETARAM France (PCTpro-E&E)

^dThe CO₂ capture experiment was performed under anhydrous conditions using a TGA/DSC thermogravimetric analyzer (10% CO₂ balanced with argon)

^eAdsorption capacity was determined by TGA based on weight gain after exposure of the dry material to a 15% CO₂/N₂

^fTGA-MS (15% CO₂, 10% H₂O, N₂ balance)

^gThe CO₂ capture was conducted in a column flow reactor system (Pure CO₂)

^hCO₂/N₂ at 15/85

ⁱ0.15 bar CO₂ and 0.75 bar N₂

^jCO₂ separation capacity, using a flow of 14% (v/v) CO₂ mixed with N₂

^kThe adsorption enthalpies, calculated from the DSC heat flow profiles during the adsorption process

^lCO₂ desorption heat determined by DSC.

^mHeat of CO₂ adsorption by DSC

New references in SI:

15. Wu, Q., Chen, S., Liu, H. Effect of surface chemistry of polyethyleneimine-grafted polypropylene fiber on its CO₂ adsorption. *RSC Adv.* **4**, 27176-27183 (2014).
16. Lin, Y., Yan, Q., Kong, C., Chen, L. Polyethyleneimine Incorporated Metal-Organic Frameworks Adsorbent for Highly Selective CO₂ Capture. *Sci. Rep.* **3**, 1859 (2013).
17. Liu, F.-Q., et al. Covalent grafting of polyethyleneimine on hydroxylated three-dimensional graphene for superior CO₂ capture. *J. Mater. Chem.* **3**, 12252-12258 (2015).
18. Zhu, J., Wu, L., Bu, Z., Jie, S., Li, B.-G. Polyethyleneimine-Grafted HKUST-Type MOF/PolyHIPE Porous Composites (PEI@PGD-H) as Highly Efficient CO₂ Adsorbents. *Ind. Eng. Chem. Res.* **58**, 4257-4266 (2019).

19. Liu, Z., et al. Moisture-resistant porous polymer from concentrated emulsion as low-cost and high-capacity sorbent for CO₂ capture. *RSC Adv.* **3**, 18849-18856 (2013).
20. Kolle, J.M., Sayari, A. Covalently Immobilized Polyethylenimine for CO₂ Adsorption. *Ind. Eng. Chem. Res.* **59**, 6944-6950 (2020).
21. Mane, S., Gao, Z.-Y., Li, Y.-X., Liu, X.-Q., Sun, L.-B. Rational Fabrication of Polyethylenimine-Linked Microbeads for Selective CO₂ Capture. *Ind. Eng. Chem. Res.* **57**, 250-258 (2018).
22. Min, K., Choi, W., Kim, C., Choi, M. Oxidation-stable amine-containing adsorbents for carbon dioxide capture. *Nat. Commun.* **9**, 726 (2018).
23. Zhou, Z., et al. Steam-Stable Covalently Bonded Polyethylenimine Modified Multiwall Carbon Nanotubes for Carbon Dioxide Capture. *Energy & Fuels* **32**, 11701-11709 (2018).
24. Kassab, H., et al. Polyethylenimine covalently grafted on mesostructured porous silica for CO₂ capture. *RSC Adv.* **2**, 2508-2516 (2012).

This new figure is mentioned in the main text as follows:

“Considering the CO₂ uptake at a low partial pressure (0.15 bar at 298 K) of any PIM base that have been reported to date, the uptake capacity of PEI-PIM-1 is one of the highest, and most certainly the most robust with the chemically tethered PEI (Supplementary Fig. 25)^{8, 14, 8, 14, 17, 18, 35, 41, 49}. Plus, PEI-PIM-1 was extended to compare with other PEI-tethered materials as shown in the Supplementary Table 10^{50, 51, 52, 53, 54, 55, 56, 57, 58, 59}.”

New references in the manuscript:

50. Wu, Q., Chen, S., Liu, H. Effect of surface chemistry of polyethyleneimine-grafted polypropylene fiber on its CO₂ adsorption. *RSC Adv.* **4**, 27176-27183 (2014).
51. Lin, Y., Yan, Q., Kong, C., Chen, L. Polyethyleneimine Incorporated Metal-Organic Frameworks Adsorbent for Highly Selective CO₂ Capture. *Sci. Rep.* **3**, 1859 (2013).
52. Liu, F.-Q., et al. Covalent grafting of polyethyleneimine on hydroxylated three-dimensional graphene for superior CO₂ capture. *J. Mater. Chem. A.* **3**, 12252-12258 (2015).
53. Zhu, J., Wu, L., Bu, Z., Jie, S., Li, B.-G. Polyethyleneimine-Grafted HKUST-Type MOF/PolyHIPE Porous Composites (PEI@PGD-H) as Highly Efficient CO₂ Adsorbents. *Ind. Eng. Chem. Res.* **58**, 4257-4266 (2019).
54. Liu, Z., et al. Moisture-resistant porous polymer from concentrated emulsion as low-cost and high-capacity sorbent for CO₂ capture. *RSC Adv.* **3**, 18849-18856 (2013).

55. Kollé, J.M., Sayari, A. Covalently Immobilized Polyethylenimine for CO₂ Adsorption. *Ind. Eng. Chem. Res.* **59**, 6944-6950 (2020).
56. Mane, S., Gao, Z.-Y., Li, Y.-X., Liu, X.-Q., Sun, L.-B. Rational Fabrication of Polyethylenimine-Linked Microbeads for Selective CO₂ Capture. *Ind. Eng. Chem. Res.* **57**, 250-258 (2018).
57. Min, K., Choi, W., Kim, C., Choi, M. Oxidation-stable amine-containing adsorbents for carbon dioxide capture. *Nat. Commun.* **9**, 726 (2018).
58. Zhou, Z., et al. Steam-Stable Covalently Bonded Polyethylenimine Modified Multiwall Carbon Nanotubes for Carbon Dioxide Capture. *Energy & Fuels* **32**, 11701-11709 (2018).
59. Kassab, H., et al. Polyethylenimine covalently grafted on mesostructured porous silica for CO₂ capture. *RSC Adv.* **2**, 2508-2516 (2012).

We thank the reviewer for the contribution.

2.7) The authors confirmed the post-synthetic modification mainly with characteristic FT-IR peaks and calculate the degree of conversion based on FT-IR peak ratios or elemental analysis which is not convincing. The NMR (carbon and nitrogen) and XPS data of each sample would be helpful to confirm the characteristic peak of introduced functional groups and further support the degree of conversion calculations.

We appreciate the reviewer for this valuable suggestion. We have now further characterized the structures with ¹³C and ¹⁵N solid state NMR to confirm the structures. In addition, we are now including XPS data of all samples and calculate the degree of conversion from the respective scans. When taking in all the additional results, we are more confident that structural verifications agreed with the previous results of FT-IR and elemental analysis. The updated results can be found in Supplementary Figure 2-3, 7, 10-11 and Supplementary Table 5: Although solid-state ¹⁵N CP-MAS NMR spectra result derived from very long experiment time could observe the important characteristic peak, we consider to exclude the spectra from the main manuscript due to the low intensity compared to the noise. We are including the ¹⁵N CP-MAS NMR spectra here for the reviewer.

Supplementary Figure 2: ^{13}C solid-state NMR spectrum of PIM-1 and derivatives.

Figure for the reviewer comments. ^{15}N solid-state NMR spectrum of PEI-PIM-1 (Top) and HZ-PIM-1 (bottom).

Supplementary Figure 3: XPS survey scan of a PIM-1, b K-PIM-1 (solvent method), c OH-PIM-1 (solvent method), d K-PIM-1, e OH-PIM-1, f MI-PIM-1, g HZ-PIM-1, and h PEI-PIM-1.

The new results are mentioned in the main text as follows:

“All chemical transformations were thoroughly assessed by FT-IR spectroscopy, elemental analysis (EA), solid-state cross-polarization magic angle spinning (CP-MAS) ^{13}C NMR, and X-ray photoelectron spectroscopy (XPS) (Fig. 3a, Table 1, and Supplementary Fig. 2-3).”

We thank the reviewer for the contribution.

2.8) On page 8, the authors stated that "The reaction between K-PIM-1 and phenylhydrazine yielded a distinct orange solution (HZ-PIM-1), exhibiting visual proof of the presence of the carbonyl group (Supplementary Fig. 3)." As the rule of thumbs, color change often implied something changed but I think it is inappropriate to include it in the scientific manuscript. If the authors think visual change proves the modifications, UV-vis spectrum analysis is recommended.

We appreciate the reviewer for his/her kind suggestion. We agree with the reviewer's viewpoint. UV-vis spectra for K-PIM-1 and HZ-PIM-1 were analyzed and attached here as a reference for the reviewer. Since we have other characterization techniques (FT-IR, XPS, and NMR) to prove the existence of ketone functionality, the color change note as a proof of conversion is now excluded from our manuscript as recommended by the reviewer.

Figure for the reviewer comments. UV-Vis spectra for K-PIM-1 and HZ-PIM-1.

Reviewer 3

Reviewer recommendation term: Minor Revision

Processable porous materials with tunable functionality are of importance for applications in molecular separations, sensors, etc. This manuscript introduces an exciting new approach for chemically modifying the polymer of intrinsic microporosity PIM-1 that can be conducted with a non-solvent and could potentially be applied, for example, in the post-treatment of membranes. I believe this will generate a great deal of interest in the polymer, porous materials and membrane communities. However, some minor corrections are required:

We truly appreciate the reviewer for their precious time in reviewing our manuscript and providing valuable comments to improve our manuscript. Below are our responses to their comments.

3.1) Line 55 “Bekir et al.” should be “Satilmis et al.” (Bekir is his given name).

We thank the reviewer for pointing this out. We revised the manuscript following reviewer’s comment. We have now corrected “Bekir et al.” to “Satilmis et al.”. Other names have also been revised accordingly.

3.2) Line 79: “called as a” would be better “called a”

We thank the reviewer for pointing this out. We revised the manuscript following reviewer’s comment. We have now corrected “called as a” to “called a”.

3.3) Line 146: “occurred to be effortless” might be better “occurred readily” or something similar

We thank the reviewer for pointing this out. We revised the manuscript following reviewer’s comment. We have now corrected “occurred to be effortless” to “occurred readily”.

3.4) Line 151: “These evidences” would be better “This evidence” (“evidence” is a body of information, “evidences” is a verb meaning to indicate clearly)

We thank the reviewer for pointing this out. We revised the manuscript following reviewer’s comment. We have now corrected “These evidences” to “This evidence”.

3.5) Line 166: “the pristine PIM-1 is incapable of reacting with amines” Although this is probably true under the conditions utilized, it might be appropriate to mention that there is the possibility of PIM-1 reacting with PEI on heat treatment, as suggested by Putintseva et al., Polymer Science, Series B, 2019, Vol. 61, No. 6, pp. 795–805.

We appreciate the reviewer for his/her constructive advice. We revised by additionally including heat treatment in the sentence as follows:

“This clearly demonstrates that without a carbonyl group, the pristine PIM-1 is incapable of

reacting with amines except if under a specific condition such as heat treatment (Supplementary Fig. 6)³⁴.”

The given reference is now cited as:

34. Putintseva, M.N., Yushkin, A.A., Bondarenko, G.N., Kirk, R.A., Budd, P.M., Volkov, A.V. Crosslinking of Polybenzodioxane PIM-1 for Improving Its Stability in Aromatic Hydrocarbons. *Polym. Sci. Ser. B* **61**, 795-805 (2019).

3.6) Line 185: “Characterizations” does not need to be plural (although one could talk about “characterization methods”)

We thank the reviewer for pointing this out. We revised the manuscript following reviewer’s comment. We have now corrected “Characterizations” to “Characterization”

3.7) Line 186: “BET Surface area isotherms using N₂ as a probe gas” This reviewer would prefer to refer to the isotherms as “N₂ adsorption/desorption isotherms” and restrict the term “BET surface area” to the apparent surface areas calculated by applying the BET equation to a portion of the isotherm.

We thank the reviewer for his/her helpful suggestion. We accordingly revised the manuscript following reviewer’s comment. We have now corrected “BET Surface area isotherms using N₂ as a probe gas” to “N₂ adsorption/desorption isotherms”.

3.8) Line 188: “Comparison of surface areas between” This reviewer would say “Comparison of N₂ adsorption/desorption isotherms and BET surface areas for”

We thank the reviewer for his/her kind advise. We revised the manuscript following reviewer’s comment. We have now corrected “Comparison of surface areas between” to “Comparison of N₂ adsorption/desorption isotherms and BET surface areas for”

3.9) Line 201: “effect of the volatility for modifying agents” Is the difference in BET surface area really due to the volatility of the reagent, and not the bulkiness of the side group that occupies free volume in the product?

We appreciate the reviewer for his/her careful observation. It is true that the bulkiness of the side group has an effect on BET surface area as the bulky group can affect morphology of the pore. However, the volatility of modifying reagent is also important for surface area preservation since it will not block the pore after chemical modification. We additionally demonstrated the significance of volatility of modifying reagent by employing the reaction between PIM-1 and tert-butylmagnesium chloride and isobutylmagnesium bromide as a representative of a bulky group. It was found that the final products derived from these two

modifying reagents could maintain surface area while phenylmagnesium bromide gave no surface area. This evidence strongly emphasizes the advantage of volatile reagent. Even the modifying agent is bulky but if it is volatile, the surface area could be preserved well after the post-modification. The results can be found in the revised Figure 3, and attached here for your convenience:

Fig. 3 | Characterization and volatile reagent effects in the non-solvent post-modification.

a FT-IR spectra and **b** N_2 adsorption/desorption isotherms measured at 77 K. **c** Comparison of N_2 adsorption/desorption isotherms and BET surface areas for post-modifications of PIM-1 by volatile and non-volatile reagents via the non-solvent method. and **d** Schematic illustration of the post-synthetic modification by using different modifying substances.

We thank the reviewer for the discussion and insightful suggestion.

3.10) Line 220-221: “A hysteresis loop was observed in all nitrogen isotherms, suggesting the presence of mesopores” The type of hysteresis observed here, which extends to low relative pressures, does NOT suggest mesopores. Mesoporosity is associated with a type of hysteresis that closes at a relative pressure of 0.4 or above. Low pressure hysteresis is characteristic of PIMs and may be attributed either to swelling of the polymer by the adsorbate or to a complex micropore structure.

We appreciate the reviewer to underline this important explanation and suggestion. To comply with the reviewer’s contribution and avoid misunderstanding, we have now rewritten this sentence as:

“All derivatives except PEI-PIM-1 showed a Type I N₂ sorption isotherm with slight behavior of a Type IV, representing combination of micro-mesoporosity^{39, 40, 41}. They had a sizeable N₂ uptake at low relative pressures. Their long hysteresis loop expanding down to low P/P₀ was also observed, which could be explained by many possible reasons such as pore network effects, the swelling effect, and diffusional limitations by pore blocking effects^{6, 42, 43}. In contrast, PEI-PIM-1 exhibited a Type IV isotherm, suggesting the transformation of micropores to mesopores. The existence of micropores and mesopores is demonstrated by the pore size distribution calculated by non-local density functional theory (NLDFT) applying the carbon slit pore model (Supplementary Fig. 24)⁴⁴.”

The following reference is now cited to convene the suggestion of the reviewer and our observations:

6. Budd, P.M., Ghanem, B.S., Makhseed, S., McKeown, N.B., Msayib, K.J., Tattershall, C.E. Polymers of intrinsic microporosity (PIMs): robust, solution-processable, organic nanoporous materials. *Chem. Commun.*, 230-231 (2004).
39. Tian, M., Rochatz, S., Fawcett, H., Burrows, A.D., Bowen, C.R., Mays, T.J. Chemical modification of the polymer of intrinsic microporosity PIM-1 for enhanced hydrogen storage. *Adsorption* **26**, 1083-1091 (2020).
40. Polak-Krasna, K., Dawson, R., Holyfield, L.T., Bowen, C.R., Burrows, A.D., Mays, T.J. Mechanical characterisation of polymer of intrinsic microporosity PIM-1 for hydrogen storage applications. *J. Mater. Sci.* **52**, 3862-3875 (2017).
41. Hu, Z.G., Wang, Y.X., Wang, X.R., Zhai, L.Z., Zhao, D. Solution-reprocessable microporous polymeric adsorbents for carbon dioxide capture. *AIChE J.* **64**, 3376-3389 (2018).
42. McKeown, N.B., Budd, P.M., Book, D. Microporous polymers as potential hydrogen storage materials. *Macromol. Rapid Commun.* **28**, 995-1002 (2007).

43. Jeromenok, J., Weber, J. Restricted Access: On the Nature of Adsorption/Desorption Hysteresis in Amorphous, Microporous Polymeric Materials. *Langmuir* **29**, 12982-12989 (2013).
44. Kupgan, G., Liyana-Arachchi, T.P., Colina, C.M. NLDFT Pore Size Distribution in Amorphous Microporous Materials. *Langmuir* **33**, 11138-11145 (2017).

3.11) Line 223: “displayed” might be better “is demonstrated by”

We thank the reviewer for pointing this out. We revised the manuscript following reviewer’s comment. We have now corrected “displayed” to “is demonstrated by”

3.12) Line 229-230: “mesopore cavity” There is no evidence of mesopores, see above comment on line 220-221. Caution should be exercised in interpreting NLDFT pore size distributions for these types of isotherms. Swelling by adsorbate may be misinterpreted as mesoporosity.

We appreciate the reviewer’s comment. We have now revised the term as “larger cavities”.

3.13) Line 233: “increase the micropores” This reviewer is not convinced the evidence is strong enough to make this claim.

We thank the reviewer for the insightful comment. We agree with the reviewer that using a volatile reagent might not simply increase micropore surface area in all cases. So, we revised this section by removing the claim.

3.14) Line 240: Table 1 “PSD” is not defined. If this is meant to indicate a pore size, this reviewer is not convinced (see comments above).

We thank the reviewers for their thoughtful comments. We agree with the reviewer that it would be better not to mention an average pore size or distribution in Table 1 since all derivatives have variety of pore sizes. We have now removed the column.

3.15) Line 334: “would be extended” should perhaps be “could be extended”

We thank the reviewer for his/her kind advise. We revised the manuscript following reviewer’s comment. We have now corrected “would be extended” to “could be extended”

OTHER NOTABLE CHANGES

- Dr. Thi Phuong Nga Nguyen was added to the author list for her contributions during this revision. She setup and measured breakthrough experiments.

- Font sizes for text in all figures adjusted to be larger for clarity.
- Updated synthesis batches of K-PIM-1 in Supplementary Table 2.
- Revised plots of pore size distribution and cumulative pore volume of PIM-1 and derivatives in Supplementary Figure 25.
- Added additional references.
- Revised funding grant numbers.

All changes were marked in highlighted text in the manuscript file. Clean versions are also provided alongside.

Reviewers' Comments:

Reviewer #1:

Remarks to the Author:

The authors have well revised the manuscript according to the reviewer comments. I am pleased to recommend its publication as is.

Reviewer #2:

Remarks to the Author:

I have carefully gone through all the reports of the reviewers and the author's point-by-point response, the authors have adequately answered most of the concerns of the reviewers. As I mentioned in the previous version of the review, I believe that this work has paved a new way to functionalize various functional groups to PIM-1 with minimum damage to porosity through the Grignard reagent. Thereby, I would like to recommend this paper be published in the nature communication journal after minor corrections. Additional explanations and characterizations are required as follows.

The CO₂ uptake capacity calculated from dynamic breakthrough experiments was not in line with the CO₂ capacity obtained from static isotherm. According to the isotherm in Figure 4b, the total CO₂ capacity of PIM-1 is less than 0.5 mmol/g at 1bar, 298 K and in the case of PEI-PIM-1, the total CO₂ capacity is less than 2.0 mmol/g at 1bar, 298 K. However, they calculated the CO₂ uptake capacity of dynamic breakthrough experiments as 1.98 mmol/g and 2.99 mmol/g for PIM-1 and PEI-PIM-1 respectively in 22 oC, 0.15 bar CO₂ partial pressure.

The capacity they calculated is much higher than the total capacity of each material at 1 bar, 273 K. Simply, it is wrong. Of course, capacity obtained from dynamic breakthrough could exhibit a small deviation between static isotherm due to calculation error or insufficient adsorption kinetics but the adsorption capacity of dynamic breakthrough at special conditions (partial pressure and temperature) "must" correspond with the adsorption capacity of static isotherm because static isotherm data were obtained by exposing the material at a specific temperature and partial pressure and wait for it till it reaches equilibrium. How can dynamic breakthrough capacity exceed the capacity obtained from equilibrium at certain conditions? The author states that " While the uptake values are significantly higher than the pure gas isotherms, the percent increase in capacities remain similar. This is in line with the mixed gas observations in the literature". But the refer 15(Qinghua Wu et al. RSC. Adv. 2014, 4, 27176) didn't compare the static isotherm and breakthrough capacity, and actually, they even reported any isotherm data in that paper they only reported dynamic breakthrough. And even if they do so, dynamic breakthrough capacity cannot exceed the equilibrium capacity.

Since the calculation equation seems right, I guess they didn't consider the blank test. To calculate the breakthrough capacity, integrated the area between the blank test breakthrough curve and the sample's breakthrough curve. And the blank test is usually conducted with non-porous glass beads with a similar amount of sample in the column. Anyway, I think the author should conduct the breakthrough test again and re-calculate the breakthrough capacity.

Reviewer #3:

Remarks to the Author:

In the revised manuscript, the authors have taken into account the comments on the originally submitted version. It is suitable for publication.

One very minor point:

p. 9, line 179: "after performed second post-modifications" would be better "after second post-modifications were performed"

Responses to Reviewer Comments

Manuscript number: NCOMMS-22-03122A

Reviewer 1

The authors have well revised the manuscript according to the reviewer comments. I am pleased to recommend its publication as is.

We deeply appreciate the reviewer as we have gained a lot from their constructive feedback to help improve the quality of our manuscript. We are also grateful to the reviewer for the positive comments and for recommending our work for publication.

Reviewer 2

I have carefully gone through all the reports of the reviewers and the author's point-by-point response, the authors have adequately answered most of the concerns of the reviewers. As I mentioned in the previous version of the review, I believe that this work has paved a new way to functionalize various functional groups to PIM-1 with minimum damage to porosity through the Grignard reagent. Thereby, I would like to recommend this paper be published in the nature communication journal after minor corrections. Additional explanations and characterizations are required as follows.

We are grateful to the reviewer for the positive comments and support in publication. We have now collected new data following the reviewer's comments. We thank for their contributions, once again.

The CO₂ uptake capacity calculated from dynamic breakthrough experiments was not in line with the CO₂ capacity obtained from static isotherm. According to the isotherm in Figure 4b, the total CO₂ capacity of PIM-1 is less than 0.5 mmol/g at 1bar, 298 K and in the case of PEI-PIM-1, the total CO₂ capacity is less than 2.0 mmol/g at 1bar, 298 K. However, they calculated the CO₂ uptake capacity of dynamic breakthrough experiments as 1.98 mmol/g and 2.99 mmol/g for PIM-1 and PEI-PIM-1 respectively in 22 °C, 0.15 bar CO₂ partial pressure.

The capacity they calculated is much higher than the total capacity of each material at 1 bar, 273 K. Simply, it is wrong. Of course, capacity obtained from dynamic breakthrough could exhibit a small deviation between static isotherm due to calculation error or insufficient adsorption kinetics but the adsorption capacity of dynamic breakthrough at special conditions (partial pressure and temperature) "must" correspond with the adsorption capacity of static isotherm because static isotherm data were obtained by exposing the material at a specific temperature and partial pressure and wait for it till it reaches equilibrium. How can dynamic breakthrough capacity exceed the capacity obtained from equilibrium at certain conditions? The author states that " While the uptake values are significantly higher than the pure gas

isotherms, the percent increase in capacities remain similar. This is in line with the mixed gas observations in the literature”. But the refer 15(Qinghua Wu et al. RSC. Adv. 2014, 4, 27176) didn't compare the static isotherm and breakthrough capacity, and actually, they even reported any isotherm data in that paper they only reported dynamic breakthrough. And even if they do so, dynamic breakthrough capacity cannot exceed the equilibrium capacity. Since the calculation equation seems right, I guess they didn't consider the blank test. To calculate the breakthrough capacity, integrated the area between the blank test breakthrough curve and the sample's breakthrough curve. And the blank test is usually conducted with non-porous glass beads with a similar amount of sample in the column. Anyway, I think the author should conduct the breakthrough test again and re-calculate the breakthrough capacity.

The reviewer's comment was highly insightful and the ensuing experiments significantly improved the quality of our manuscript. We thank them for that. We have now carried out the dynamic breakthrough experiments (in three different systems, then selected the most reproducible one) including the blank test, from which the dead volume was derived. The results were reported briefly in the main text, revising the corresponding section. The details of CO₂ breakthrough experiments and calculations can now be found in Supplementary Information section 5. The breakthrough curves of PIM- and PEI-PIM-1 were updated in the new Supplementary Figure 38-39. We will also report them here for the convenience of the reviewer:

First, detailed methods:

5. CO₂ Breakthrough Experiment

5.1 Methodology

Well-ground polymer was filled into a stainless-steel column and activated at 120 °C for 5 hours prior to breakthrough adsorption test. Before each experiment, helium reference gas was flushed through the column and the gas flow was then switched to the desired gas mixture at the flow rate of 5 mL/min (80.75% N₂, 14.25% CO₂, 5% He). The test was carried out at 1 bar pressure and 25 °C. Outcoming gases were analyzed by a gas analysis system (QGA, Hiden analytical, United Kingdom). The blank tests were conducted with empty column to suppress effect of system configuration on adsorption calculation.

Schematic diagram of the column breakthrough experiment used in this study.

5.2 Derivation of adsorption capacity

Adsorption capacity is calculated as the amount of uptake adsorbate per unit mass of the adsorbent.

$$q_t = \frac{m_{ad}}{m} \quad (1)$$

While,

$$m_{ad} = C_{ad}V \quad (2)$$

$$V = Ft \quad (3)$$

$$C_{ad} = C_o - C_t \quad (4)$$

Substitute (2), (3) and (4) into (1),

$$q_t = \frac{F}{m} (C_o - C_t)t \quad (5)$$

Then total adsorbate will be calculated by Eq. (6)

$$q = \frac{F}{m} \int_0^{t_s} (C_o - C_t) dt$$

$$\text{or } q = \frac{FC_o}{m} \int_0^{t_s} \left(1 - \frac{C_t}{C_o}\right) dt, \quad (6)$$

Where,

q : Adsorption capacity (mmol/g)

q_t : Adsorption capacity at time t (mmol/g)

m_{ad} : Mass of adsorbate (g)

m : Mass of adsorbent (g)

C_{ad} : Concentration of adsorbate taken by adsorbent (g)

V : Volume of adsorbate (L)

F : Flow rate of adsorbate (L/min)

t : adsorption time (min)

C_o : Initial concentration of adsorbate (mmol/L)

C_t : Concentration of adsorbate come out from fixed bed (mg/L)

t_s : Saturated time (min)

5.3 Calculation from experimental data:

$$F = 5 \text{ mL/min CO}_2 = 0.005 \text{ L/min}$$

$$C_o = \frac{1}{24.76\text{L}} \times 14.25\% = 5.755 \text{ mmol/L}$$

(At 1 bar, 25 °C, 1 mole of ideal gas is 24.76 L)

$m = 181 \text{ mg}$ of PIM-1 and 236 mg PEI-PIM-1

$\int_0^{t_s} \left(1 - \frac{C_t}{C_o}\right) dt$ is the subtracted area between the plot $\frac{C}{C_o} = 1$ and $\frac{C}{C_o} = \frac{C_t}{C_o}$ versus time t

$$\text{For blank test, } \int_0^{t_s} \left(1 - \frac{C_t}{C_o}\right) dt = 1.253$$

$$\text{For PIM-1, } \int_0^{t_s} \left(1 - \frac{C_t}{C_o}\right) dt = 3.228$$

Then adsorption capacity (q) of PIM-1 is 0.31 mmol/g

$$\text{For PEI-PIM-1 } \int_0^{t_s} \left(1 - \frac{C_t}{C_o}\right) dt = 6.011$$

Then adsorption capacity (q) of PEI-PIM-1 is 0.58 mmol/g

Figure for the response file. Blank runs (80.75% N_2 , 14.25% CO_2 , 5% He mixture at 1 bar and 298K.)

Supplementary Figure 38: Breakthrough curves of PIM-1 using $N_2/CO_2/He$ (80.75/14.25/5) gas mixtures at 1 bar and 298 K.

Supplementary Figure 39: Breakthrough curves of PEI-PIM-1 using N₂/CO₂/He (80.75/14.25/5) gas mixtures at 1 bar and 298 K.

The main text was modified accordingly as follows:

“To further appraise the performance of PEI-PIM-1 compared with PIM-1 in the actual adsorption-based separation process, CO₂ breakthrough experiments for PEI-PIM-1 and PIM-1 were tested at ambient temperature using 80.75% N₂, 14.25% CO₂, 5% He gas mixtures (Supplementary Fig. 38-39). The dynamic adsorption capacity of PEI-PIM-1 (0.58 mmol/g) was superior to PIM-1 (0.31 mmol/g), which is consistent with the pure gas isotherms (see Supplementary Information section 5 for methods and calculations). The adsorption value obtained from static and dynamic conditions may not be equivalent as it is theoretically known that the former yields higher CO₂ uptake than the latter^{68, 69}. Also, axial dispersion, mass transfer and adsorption kinetics effects take place during the dynamic operation, influencing the shape of breakthrough curve used for calculating the adsorption capacity⁷⁰. Apart from the adsorption capacity, it is noticeable that CO₂ breakthrough curve of PEI-PIM-1 showed wider breakthrough time gap between CO₂ and N₂ compared to the parent PIM-1, indicating a substantially better separation performance of CO₂/N₂ gas pair.

Additional references in the manuscript:

69. Tiwari, D., Goel, C., Bhunia, H. & Bajpai, P. K. Dynamic CO₂ capture by carbon adsorbents: Kinetics, isotherm and thermodynamic studies. *Sep. Purif. Technol.* **181**, 107-122 (2017).

70. Bhatt, P. M. et al. A Fine-Tuned Fluorinated MOF Addresses the Needs for Trace CO₂ Removal and Air Capture Using Physisorption. *J. Am. Chem. Soc.* **138**, 9301-9307 (2016).

71. García, S. et al. Breakthrough adsorption study of a commercial activated carbon for pre-combustion CO₂ capture. *J. Chem. Eng.* **171**, 549-556 (2011).

We thank the reviewer for the insightful contribution.

Reviewer 3

In the revised manuscript, the authors have taken into account the comments on the originally submitted version. It is suitable for publication.

One very minor point:

p. 9, line 179: "after performed second post-modifications" would be better "after second post-modifications were performed"

We revised the manuscript following reviewer's comment by rewriting the sentence from "after performed second post-modifications" to "after second post-modifications were performed".

Once again, we deeply appreciate the reviewer for all their contributions to improve our manuscript and for acknowledging that this work is appropriate for publication in Nature Communications.

OTHER NOTABLE CHANGES

- Abdulhadi Alhaji, Mohamed Eddaoudi, and Mert Atilhan were added to the author list for their contributions during this revision. In author contributions, we added them as "...A. A. and M. E. setup and analyzed breakthrough data. M. A. contributed to the discussion of breakthrough data."

All changes were marked in highlighted text in the manuscript file. Clean versions of the manuscript files are also provided for convenience.

Reviewers' Comments:

Reviewer #2:

Remarks to the Author:

The authors made further corrections and this manuscript could be accepted as is.

Responses to Reviewer Comments
Manuscript number: NCOMMS-22-03122B

Reviewer 2

Reviewer recommendation term: Accepted

The authors made further corrections and this manuscript could be accepted as is.

We deeply appreciate the reviewer for his/her valuable time and effort to review this manuscript. His/her insightful comments and suggestions help us improve the quality of our manuscript greatly. We thank the reviewer again for the acceptance of this work to publish in Nature Communications.

OTHER NOTABLE CHANGES

- The manuscript has been revised according to editorial comments as indicated in the checklist.

All changes were marked in highlighted text in the manuscript file. Clean versions are also provided alongside.